# Vertex protein PduN tunes encapsulated pathway performance by dictating bacterial metabolosome morphology

Carolyn E. Mills[1], Curt Waltmann[2], Andre G. Archer[3], Nolan W. Kennedy[4], Charlotte H. Abrahamson[1], Alexander D. Jackson[5], Eric W. Roth[6], Sasha Shirman[3], Michael C. Jewett [1,7], Niall M. Mangan[3,4,7], Monica Olvera de la Cruz [2,7,8] & Danielle Tullman-Ercek [1,7✉]

Engineering subcellular organization in microbes shows great promise in addressing bottle-necks in metabolic engineering efforts; however, rules guiding selection of an organization strategy or platform are lacking. Here, we study compartment morphology as a factor in mediating encapsulated pathway performance. Using the 1,2-propanediol utilization micro-compartment (Pdu MCP) system from *Salmonella enterica* serovar Typhimurium LT2, we find that we can shift the morphology of this protein nanoreactor from polyhedral to tubular by removing vertex protein PduN. Analysis of the metabolic function between these Pdu microtubes (MTs) shows that they provide a diffusional barrier capable of shielding the cytosol from a toxic pathway intermediate, similar to native MCPs. However, kinetic modeling suggests that the different surface area to volume ratios of MCP and MT structures alters encapsulated pathway performance. Finally, we report a microscopy-based assay that permits rapid assessment of Pdu MT formation to enable future engineering efforts on these structures.

[1] Department of Chemical and Biological Engineering, Northwestern University, Evanston, IL, USA. [2] Department of Materials Science and Engineering, Northwestern University, Evanston, IL, USA. [3] Department of Engineering Sciences and Applied Mathematics, Northwestern University, Evanston, IL, USA. [4] Interdisciplinary Biological Sciences Program, Northwestern University, Evanston, IL, USA. [5] Master of Science in Biotechnology Program, Northwestern University, Evanston, IL, USA. [6] Northwestern University Atomic and Nanoscale Characterization Experimental Center, Evanston, IL, USA. [7] Center for Synthetic Biology, Northwestern University, Evanston, IL, USA. [8] Department of Chemistry, Northwestern University, Evanston, IL, USA. ✉email: ercek@ northwestern.edu

Spatial organization of biological processes is essential to life across many organisms, from multicellular eukaryotes to unicellular prokaryotes. Once thought to lack subcellular organization, bacteria utilize an array of strategies for segregating specific processes within the cell. One such example is bacterial microcompartments (MCPs), which are organelles that encase specific sets of enzymes in a protein shell[1,2]. Genes associated with MCPs are found in 45 bacterial phyla[3,4], and are classified by the metabolic pathway segments they encapsulate. At the highest level, MCPs are classified as either carboxysomes or metabolosomes based on whether they encase pathways involved in anabolic or catabolic processes, respectively[1]. Carboxysomes aid many carbon-fixing bacteria by increasing $CO_2$ concentration in the vicinity of the carboxylating enzyme ribulose bisphosphate carboxylase/oxygenase (RuBisCO)[5,6]. Metabolosomes, on the other hand, aid in metabolism of a broad array of substrates and thus encapsulate many different pathway chemistries; however, these pathways typically share a unifying feature of passing through a toxic aldehyde intermediate[7,8]. Sequestration of this toxic intermediate is thought to aid in metabolism of niche carbon sources such as 1,2-propanediol and ethanolamine, providing a competitive growth advantage to the enteric pathogens that often harbor metabolosomes[9,10].

MCPs represent attractive engineering targets in a variety of applications, from bioproduction, where heterologous enzyme encapsulation could improve pathway performance[11], to antibiotic development, where disruption of these MCP structures could eliminate a competitive growth advantage[9]. However, metabolosomes in particular exhibit diversity in shape and size, and it is not well-understood how these features relate to function[4,12–15]. A variety of engineering fields, from catalysis[16] to drug delivery[17], have illustrated the importance of shape and size on nanomaterial performance. The relevance of these features has yet to be meaningfully investigated in MCP systems.

The 1,2-propanediol utilization (Pdu) MCP is a model metabolosome that aids in breakdown of 1,2-propanediol[18]. Pdu MCPs exist in a variety of bacteria[3,4,10], and both the encapsulated pathway[10,18,19] and the structure[20] of these metabolosomes have been investigated. The *pdu* operon contains 21 genes encoding for the proteins that make up the Pdu MCP shell as well as the main pathway and cofactor recycling enzymes (Fig. 1). Eight proteins compose the Pdu microcompartment (MCP) shell —PduA, PduB, PduB', PduJ, PduK, PduN, PduT, and PduU[21,22]. Of these eight proteins, seven (PduABB'JKTU) contain one or more bacterial microcompartment (BMC) pfam00936 domains, and, as such, form the hexagonal multimers that assemble into the facets and edges of the microcompartment[22–27]. *pduN* is the sole bacterial microcompartment vertex (BMV) pfam03319 gene in the *pdu* operon and is thus expected to form pentamers that cap the vertices of the Pdu MCP[15,28–32]. PduN is a low abundance component of the MCP shell, but it is essential for the formation of well-formed compartment structures[21,22,33]. While prior studies have illustrated that aberrant structures form in the absence of PduN, the functionality and nature of these structures have yet to be explored in any detail. Further, studies on both alpha- and beta-carboxysomes showed that strict closure of the shell is required for these microcompartments to confer their biologically relevant growth benefits and that this cannot be achieved in the absence of pentameric vertex shell proteins like PduN[34,35]. It is unclear how important this strict closure is for metabolosome systems like the Pdu MCP, as modeling studies have shown that a moderate diffusional barrier between the cytosol and an enzyme core is sufficient for mediating toxic intermediate buildup[36]. Previous work has suggested the differing importance of various shell proteins, including PduN, in Pdu MCP function[22]; but questions remain about precisely how MCP morphology controls Pdu pathway performance.

Here, we describe our detailed characterization of an MCP-related structure that we call Pdu microtubes (Pdu MTs) that form when vertex protein PduN cannot incorporate into the Pdu MCP shell, and use molecular dynamics modeling to understand the molecular interactions responsible for this morphology shift. We investigate how encapsulated pathway performance is impacted by this shift in morphology and use kinetic modeling to interrogate what features of compartment geometry control toxic intermediate buildup in the cell. Finally, we present a microscopy-based assay that screens for formation of these Pdu MTs, enabling investigation of the key molecular features that govern PduN incorporation into the Pdu MCP. Together, these results represent a key step towards understanding the complex interplay between shell protein interactions, compartment morphology, and encapsulated pathway performance.

## Results

**PduN mediates the morphology of Pdu compartment structures.** We first explored the impact of PduN on in vivo assembly

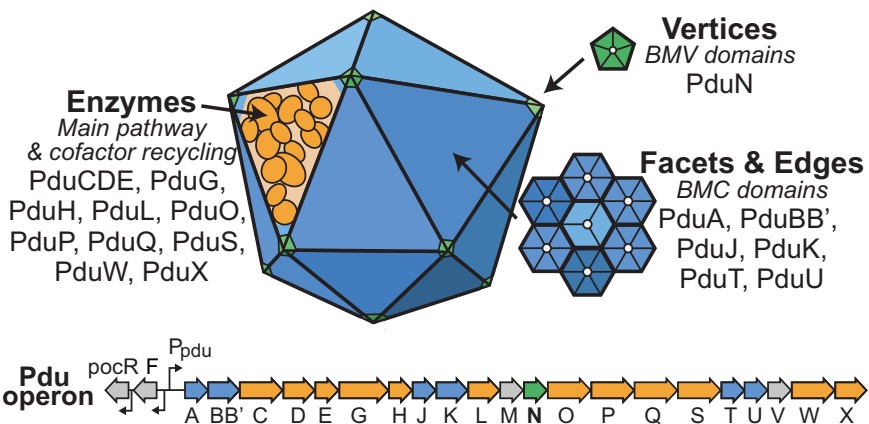

**Fig. 1 The 1,2-propanediol utilization microcompartment from *Salmonella* is made up of proteins encoded by the *pdu* operon.** The *pdu* operon in *Salmonella enterica* serovar Typhimurium LT2 contains the genes encoding proteins responsible for formation of the 1,2-propanediol utilization microcompartment (Pdu MCP). These include enzymes that perform both key pathway steps and cofactor recycling functions (orange) and shell proteins that encase these enzymes (bacterial microcompartment, BMC, domain-containing genes shown in blue, bacterial microcompartment vertex, BMV, domain-containing gene shown in green). Notably, only one shell protein in the *pdu* operon, PduN, contains a BMV domain.

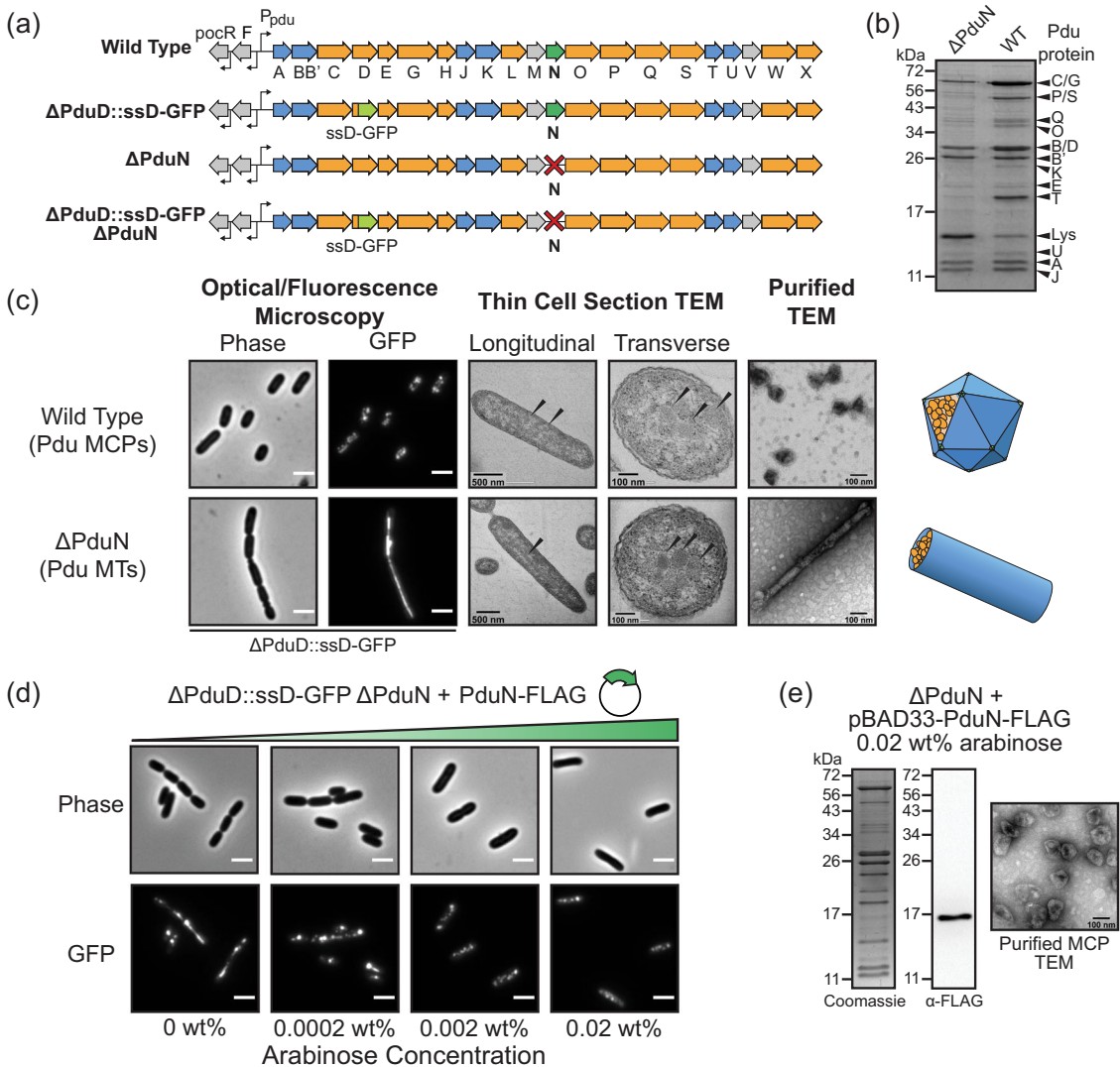

**Fig. 2 Characterization of structures formed in the absence of PduN. a** Depiction of different *pdu* operon genotypes used in this figure. **b** Coomassie-stained SDS-PAGE of purified Pdu MTs (ΔPduN) and Pdu MCPs (WT) comparing the protein content in these purified structures, where labels indicate the Pdu protein the band corresponds to (i.e. C/G for PduC/PduG), except for Lys, which indicates lysozyme. **c** Comparison of structures formed in Pdu MCP-forming strains (WT) and Pdu MT-forming strains (ΔPduN). Scale bars in optical and fluorescence micrographs are 5 μm. **d** Phase contrast and GFP fluorescence micrographs showing the impact of increased PduN-FLAG expression on the formation of Pdu MT structures versus closed Pdu MCP structures, where increasing arabinose concentration correlates with increasing expression of the PduN-FLAG protein off the pBAD33 plasmid. Scale bars in optical and fluorescence micrographs are 5 μm. **e** Coomassie-stained SDS-PAGE, anti-FLAG western blot, and negatively stained TEM on Pdu MCPs purified from a *pduN* knockout strain supplemented with PduN-FLAG off a plasmid. Source data for (**b**) and (**e**) are provided as a Source Data file. Similar results to those reported in (**b**–**e**) were observed across three independent biological replicates, except for TEM imaging of thin cell sections, which was performed on multiple cells in a given biological sample, but not with biological replicates.

of Pdu MCPs using a combination of fluorescence microscopy and transmission electron microscopy (TEM) on thin cell sections of both wild type (WT, PduN-containing) and *pduN* knockout strains (ΔPduN). Our fluorescence microscopy assay uses a green fluorescent protein (GFP) reporter fused to an encapsulation peptide, herein referred to as ssD for signal sequence from PduD, that is sufficient for encapsulation of heterologous proteins in Pdu MCPs[37]. Thus, compartment distribution throughout the cell is indicated by the presence of the green fluorescence associated with the ssD-GFP reporter encapsulated within the MCP lumen. As in previous studies, expression of Pdu MCPs in the wild type (PduN-containing) background results in punctate fluorescence throughout the cell (Fig. 2c), suggesting that well-formed compartments are distributed within the cell[38–40]. In contrast, when the *pduN* gene is knocked out,

expression of the *pdu* operon results in lines of fluorescence, typically aligned with the long axis of the cell (Fig. 2c). These lines of fluorescence indicate the formation of elongated structures within the cell capable of recruiting ssD-tagged GFP. Indeed, thin cell section TEM on cells expressing the *pdu* operon in the *pduN* knockout strain confirms the presence of tube structures, henceforth referred to as Pdu MTs (Fig. 2c). Interestingly, both fluorescence microscopy and thin cell section TEM show that these Pdu MTs appear to inhibit cell division, as the structures traverse multiple cleavage furrows (Fig. 2c, Supplementary Fig. 1). While striking, such elongated structures are not unprecedented in the MCP literature—similar extended structures have also been observed in cells expressing pentamer-deficient carboxysomes, for example[34]. However, little is known about the structure or protein content of these tube structures.

We thus sought to examine, in detail, the structure of the Pdu MTs formed by expression of the *pdu* operon in our *pduN* knockout strain. These Pdu MTs are comprised of many of the same shell proteins as Pdu MCPs, evidenced by the presence of PduA, PduB, PduB', PduJ, and PduU bands by SDS-PAGE in both samples (Fig. 2b). Notably, bands associated with enzymatic cargo (PduCDE, PduG, PduP, PduQ, PduS) are also present in the purified Pdu MT sample. TEM analysis of purified Pdu MTs (Fig. 2c, Supplementary Fig. 2) shows that these tubes are 50 ± 10 nm in diameter, in agreement with diameters observed in cell sections. This dimension is distinct from the 20 nm diameter of rods self-assembled from PduA and PduJ shell proteins alone[23,41], indicating that some combination of the other shell proteins present and the encapsulated cargo mediates the size and curvature of these Pdu MTs[41,42]. These results suggest that the Pdu MTs formed by our *pduN* knockout strain are complex multi-protein assemblies, similar to Pdu MCPs.

Observing that knocking out *pduN* caused the formation of Pdu MTs instead of Pdu MCPs, we hypothesized that PduN is directly responsible for mediating the morphology of Pdu microcompartments. To test this hypothesis, we supplemented our *pduN* knockout strain with a plasmid containing FLAG-tagged PduN and observed changes in compartment morphologies at varying inducer levels using fluorescence microscopy (Fig. 2d). We find that increasing PduN-FLAG expression decreases the formation of elongated structures (Pdu MTs), and increases the observation of punctate fluorescence (Pdu MCPs) (Fig. 2d). Interestingly, even with no inducer present (0 wt% arabinose), a decrease in the percent of cells with elongated structures is observed (Supplementary Fig. 3). This is likely a result of leaky PduN-FLAG expression; because PduN constitutes only 0.6% of the total shell protein content, it is not surprising that even very low levels of PduN would impact shell closure[43]. We validated these microscopy results by purifying compartments from a *pduN* knockout strain supplemented with PduN-FLAG off a plasmid (0.02 wt% arabinose). These compartments exhibit the characteristic polyhedral geometry of Pdu MCPs by TEM and the characteristic banding pattern of well-formed Pdu MCPs by SDS-PAGE (Fig. 2e). Further, anti-FLAG western blotting on these same purified compartments confirmed the presence of PduN-FLAG in these well-formed structures (Fig. 2e). We conclude that PduN plays a direct role in the formation of Pdu MCPs, likely by facilitating capping of MCP vertices.

Next, we examined the molecular underpinnings of how PduN facilitates MCP closure by investigating the interaction interface responsible for PduN incorporation using all-atom molecular dynamics (AAMD) simulations. Previous work modeling the interface between two PduA hexamers revealed that preferred interaction angles between hexamers play a key role in higher-order assembly of these proteins[44]. We hypothesized that similar studies comparing the PduN interaction interface to the PduA/PduA interface could yield insight into the specific, unique features that allow PduN to initiate Pdu MCP vertex capping. We selected PduA as the interacting partner for PduN based on previous studies showing that PduA and PduN interact ex vivo[21]. We built an estimated model of the PduA/PduN and PduA/PduA interfaces using a homology-based approach that leveraged the solved crystal structure of the HO MCP (PDB: 5V74). This structure provides exquisite molecular detail of how homologous shell proteins assemble to form an MCP shell (see Methods for details)[15,42]. Using this model as a starting point, we performed AAMD simulations of this interface to examine the energetics associated with various bending angles between PduA and PduA as well as between PduA and PduA (Fig. 3). Specifically, we calculated the potential of mean force (PMF) between each pair of protein oligomers as a function of the bending angle between the

two components (Fig. 3b, e) and the distance between their centers of mass in the y-direction (Fig. 3c). More details on the calculation can be found in the Methods Section and Supplementary Methods. The resulting PduA/PduN bending energy landscape revealed a strong preference for a 40° bending angle between PduA and PduN (Fig. 3b) with the bending energy ($\Delta G_{0° \rightarrow 40°} = -6 \pm 2$ kcal/mol) comprising over half of the total interaction strength ($\Delta G_{PduN/PduA} = -10 \pm 2$ kcal/mol, Fig. 3c). Notably, this is higher than bending angles (30°) between hexamer/pentamer components reported in the crystal structure of an MCP shell from *Haliangium ochraceum*[15]. This preference for a bent interaction is distinct from the bending energy landscape of the PduA/PduA interface, which has only shallow minima ($\Delta G_{0° \rightarrow 34°} = -1.2 \pm 0.3$ kcal/mol) that constitute less than a quarter of the total PduA/PduA interaction energy ($\Delta G_{PduA/PduA} = -11 \pm 2$ kcal/mol[44]) We note two things about this bending energy landscape. First, that the energy minimum at 34° is consistent with previous models investigating PduA/PduA bending interactions[41]. Second, we note that a second minimum exists at a PduA/PduA interaction angle of ~70°; however, given that this bending angle would not permit assembly of larger icosahedra or polyhedra like those formed in the Pdu MCP system, we do not believe it is physically relevant to the discussion here. Interestingly, while the bending angle preference is dramatically different between these two interfaces, the magnitude of the PduA/PduN and PduA/PduA interaction is similar[44] (Fig. 3c). Together, this suggests that PduN could provide an energetically favorable bending point that allows for the closure of the shell without requiring less favorable bending of the PduA/PduA interface. Since this bending is intrinsic to the PduA/PduN interaction, even dimers, trimers, or any other PduN-featuring oligomers would also be highly bent. Thus, their incorporation would quickly disrupt the formation of any smaller Pdu MTs or flat sheets that are likely present early in the assembly process due to the low concentration of PduN.

**Pdu microtubes control the metabolic flux of the 1,2-propanediol utilization pathway.** Having shown that elongated Pdu MTs form in the absence of PduN, we next sought to probe the metabolic functionality of these tubes, and how organization into MTs versus MCPs impacts pathway performance. We hypothesized that the morphological shift from Pdu MCPs to MTs may negatively impact pathway performance, as we expect the Pdu MTs to have open ends that would increase exchange between the enzymatic core and the cytosol.

We explored the impact of compartment geometry on Pdu pathway performance by comparing the growth and external Pdu metabolite profiles (1,2-propanediol, propionaldehyde, 1-propanol, and propionate, Fig. 4a) of four strains—wild type (MCP-forming), ΔPduN (MT-forming), ΔPduA PduJ (broken compartment control[23]), and ΔPocR (no *pdu* operon expression control[45–47]). We grew these strains on 1,2-propanediol with excess adenosylcobalamin (adoB12), a condition that permits distinction of compartment-forming conditions that successfully sequester the toxic propionaldehyde intermediate away from the cytosol[22,48,49]. Cell growth and metabolite profiles (Fig. 4b, c) show that control strains, ΔPocR and ΔPduA PduJ, grow as expected. When there is no expression of the *pdu* operon (ΔPocR), there is no cell growth over time, as none of the enzymes capable of 1,2-propanediol metabolism are present (Fig. 4b). Metabolite tracking confirms that no 1,2-propanediol is consumed (Fig. 4c). When the operon is expressed, but compartments cannot properly form (ΔPduA PduJ), cell growth and 1,2-propanediol consumption initially occur rapidly (Fig. 4b, c; doubling time of 2.338 ± 0.003 h between 3 and 9 h), as there is

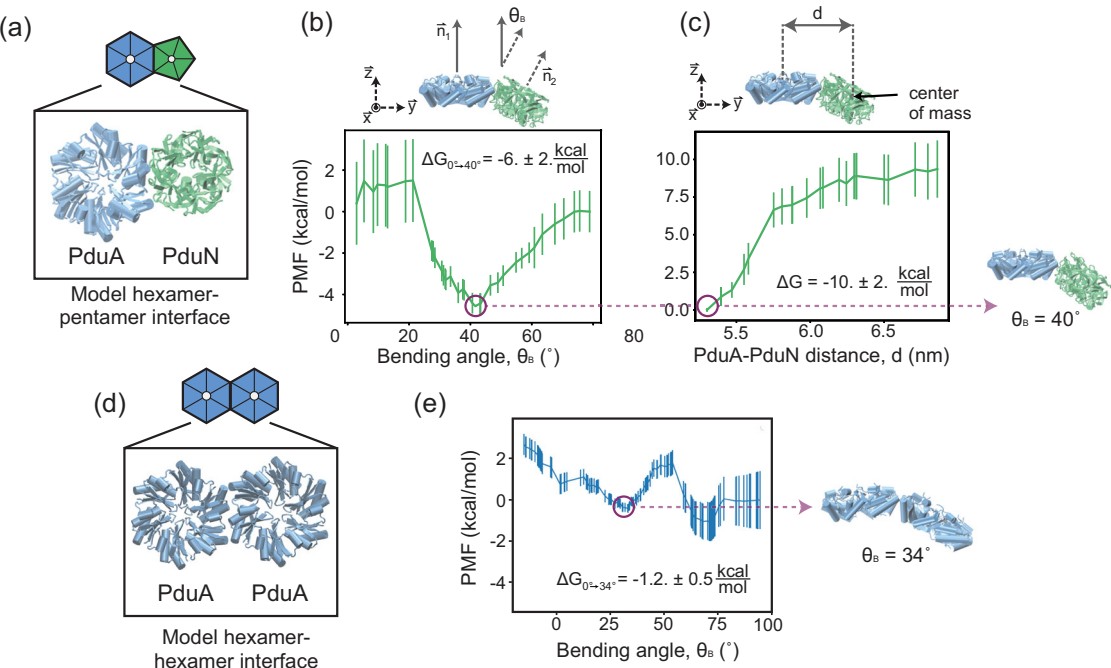

**Fig. 3 Analysis of the molecular interactions responsible for PduN incorporation into the shell using all-atom molecular dynamics (AAMD) simulations. a** Schematic of the PduA-PduN interface used for these simulations, where PduA is shown in blue and PduN is shown in green. **b** Potential of mean force (PMF) calculated from AAMD simulations as a function of bending angle, $\Theta_B$, between PduA and PduN. $\Delta G_{0°→40°}$ is the difference in the PMF between the 0° and 40° bending angles. **c** PMF calculated from AAMD simulations as a function of the distance between PduA and PduN, used to calculate the total interaction energy ($\Delta G$) between these two oligomers. **d** Schematic of the PduA-PduA interface used for these simulations. **e** PMF calculated from AAMD simulations as a function of bending angle, $\Theta_B$, between two PduA hexamers. $\Delta G_{0°→34°}$ is the difference in the PMF between the 0° and 34° bending angles. Calculations used calculate data points in (**b**), (**c**), and (**e**) are described in Supplementary Method 1. Error bars on plots in (**b**), (**c**), and (**e**) represent the sampling error on the calculated energies, estimated by splitting simulation data into different sections and observing the differences in the calculated potential as described in Supplementary Method 1, Calculation specifics. Source data for plots (**b**), (**c**), and (**e**) are provided as a Source Data file.

no shell protein barrier preventing enzymes access to 1,2-propanediol. Consequently, this strain exhibits the most rapid initial generation of propionaldehyde, propionate and 1-propanol (Fig. 4c). However, after ~12 h, a lag in growth begins to occur as propionaldehyde buildup exceeds a threshold value (doubling time of $62 ± 18$ h between 12 and 18 h). This stalls propionate uptake into central metabolism, explaining both the observed growth lag and the delayed propionate consumption in this strain between 12 and 30 h (Fig. 4b). Several groups have reported this in strains with a broken compartment phenotype[19,22,23,49], where it was hypothesized that propionaldehyde inhibits the methylcitrate pathway[50].

In contrast, strains containing Pdu MCPs (wild type) and Pdu MTs (ΔPduN) exhibit growth profiles consistent with a well-encapsulated Pdu pathway[22]. Initial growth and 1,2-propanediol consumption are slightly slower than the broken compartment control (ΔPduA PduJ), corresponding to doubling times of $3.17 ± 0.08$ h and $2.64 ± 0.13$ for wild type and ΔPduN strains between 3 and 9 h, respectively. However, growth of WT and ΔPduN strains eventually surpass the ΔPduA PduJ strain at later time points as propionaldehyde buildup begins to impact growth, evidenced by doubling times of $9.2 ± 1.6$ h for the WT strain and $9.2 ± 1.9$ h for the ΔPduN strain between 12 and 18 h. Strains containing Pdu MCPs (WT) and Pdu MTs (ΔPduN) both exhibit a lower peak concentration of propionaldehyde than the broken compartment strain (ΔPduA PduJ); however, the buildup of propionaldehyde is slightly more rapid in the Pdu MT strain than the Pdu MCP strain, where there are detectable propionaldehyde levels at 9 h of growth (Fig. 4c). This suggests that the change in geometry from MCP to MT subtly alters passive substrate transport in and out of the compartment, impacting the accessibility of substrates to the enzymatic core. This could either be due to changes in compartment surface area or potential open ends of Pdu MTs. Significantly, compared to the Pdu MCP strain (WT), the Pdu MT strain (ΔPduN) exhibits lower peak propionate concentrations and more rapid consumption of 1-propanol (Fig. 4c). This suggests that in these growth conditions, the Pdu MT geometry favors more rapid uptake of propionate into central metabolism, again, possibly due to changes in average substrate transport in and out of Pdu MTs versus Pdu MCPs. Taken together, these results indicate that the diffusional barrier provided by the Pdu MT protein shell is sufficient to prevent toxic propionaldehyde buildup in the cytosol.

**Systems-level kinetic modeling shows how compartment geometries impact pathway flux.** Observing that strains containing Pdu MCPs and Pdu MTs exhibited slightly different metabolite profiles over time, we hypothesized that the key difference between the encapsulation vehicles in question, MCPs and MTs, is the surface area to volume ratio of these structures. To interrogate this hypothesis, we modified a systems-level kinetic model of the Pdu pathway[36] to account for cell growth and treat the MCP and MT geometries. We parameterized our model using literature values, and then adjusted the MCP permeability and PduP/PduQ concentration to match features of propionaldehyde time evolution in external media for the wild type strain (see Supplementary Table 1). Specifically, MCP permeability was adjusted such that the timescale of propionaldehyde buildup and degradation matched that observed in experiments, and PduP/PduQ activities were adjusted such that the maximum

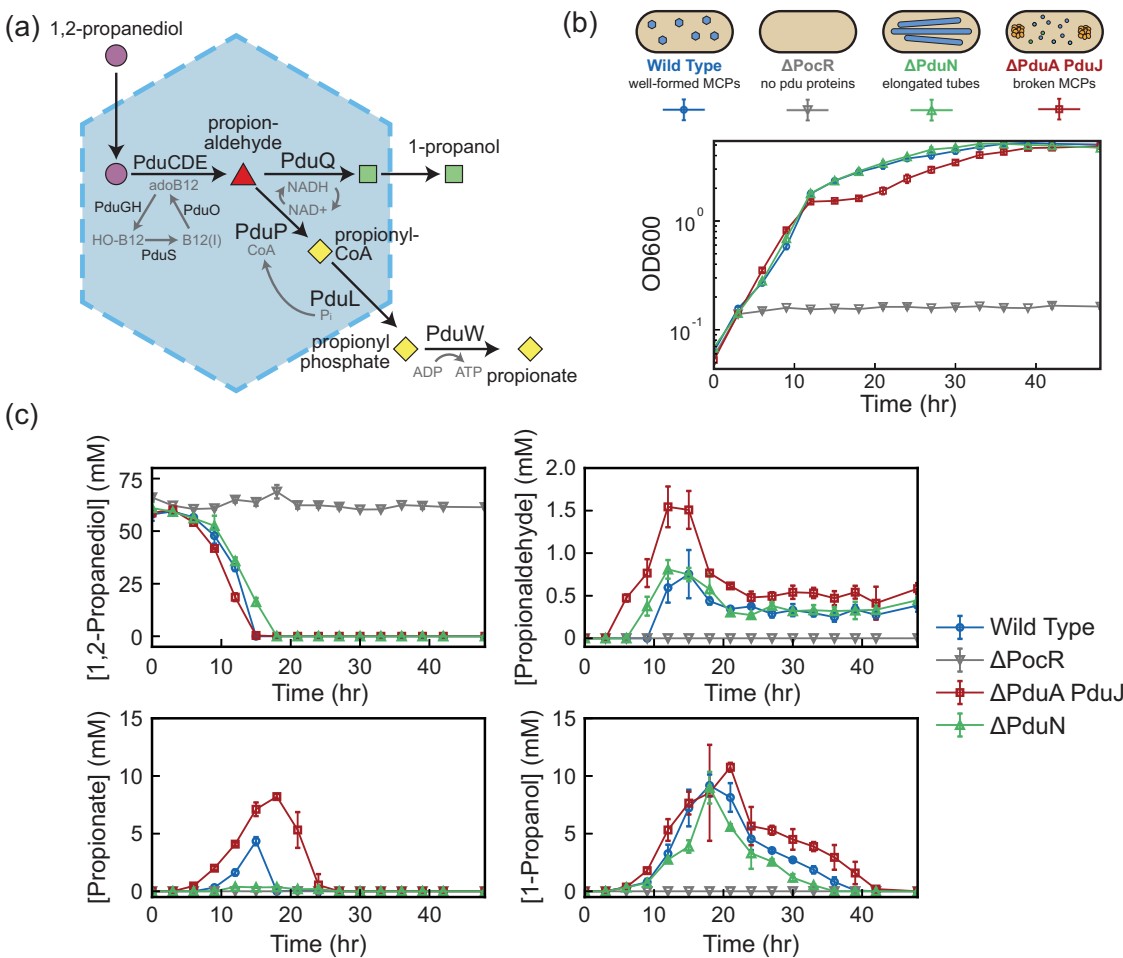

**Fig. 4 Impact of compartment geometry on 1,2-propanediol utilization pathway performance. a** Schematic of the 1,2-propanediol utilization pathway encapsulated in Pdu microcompartments. **b** Strains containing different compartment geometries (MCPs in Wild Type, blue lines, MTs in ΔPduN, green lines), without compartment expression (ΔPocR, grey lines), and with broken compartments (ΔPduA PduJ, red lines) grown in minimal media (NCE) with 1,2-propanediol as the sole carbon source. Data are presented as mean values ± standard deviation over three biological replicates. **c** Concentration of key pathway metabolites over the course of the growth described in (**b**). Data are presented as mean values ± standard deviation over three biological replicates. Source Data for panels (**b**) and (**c**) are provided as a Source Data file.

propionaldehyde level did not exceed levels previously reported to cause dramatic growth defects (<16 mM)[19]. Pdu MCPs are modeled as spheres 140 nm in diameter[20] with 15 MCPs per cell, where MCP permeability controls access to enzymes encapsulated in the MCP (further model details are provided in Supplementary Method 2). The timescales of extracellular 1,2-propanediol consumption and 1-propanol/propionate buildup in this Pdu MCP model (Fig. 5c) match our experimental data (Fig. 4c) well—1,2-propanediol consumption occurs between 10 and 20 h, and initial propionate and 1-propanol buildup is observed at 10 h. This suggests that our model parameterization has correctly captured key features of dynamic Pdu pathway behavior. We note two discrepancies between the model and our experimental data: (1) propionate and 1-propanol are eventually consumed in our experiments, and (2) absolute propionaldehyde concentrations observed differ from those predicted in the model. These differences are primarily due to exclusion of downstream reactions from the model and propionaldehyde volatility and reactivity under experimental conditions, and are discussed in detail in the Supplementary Discussion 1. Given that the timescales of propionaldehyde buildup and consumption match our model, we believe that our model is sufficiently accurate to allow comparison of propionaldehyde buildup in different compartment geometries.

We next adjusted our model to study how changing the geometry of the compartment from spherical (MCP) to cylindrical (MT) impacted pathway performance. We assume that Pdu MTs are cylinders 50 nm in diameter, and length equal to the length of the cell (2.5 μm). Metabolites diffuse into MTs through the surface along the long axis of the cylinder, but not the ends (see Supplementary Method 2 for further modeling details and Supplementary Discussion 1 for discussion of diffusion out the ends of the cylinder). The final parameter to set in the model, then, was the number of Pdu MTs per cell. To allow direct comparison to the spherical MCP base case, we kept as many parameters as possible equivalent in the MT model, including enzyme number. We then tested the effect of differences in geometry between the Pdu MT and MCP cases, namely a change in the surface area to volume ratios between these two geometries. We could keep either the total surface area or the encapsulated volume constant, but not both. We thus considered two limiting cases: (1) total compartment surface area is the same in the MCP and MT models and enzyme concentration increases or (2) total volume is the same in the MCP and MT models and the enzyme concentration is constant. Comparison of these two limiting cases illustrates the substantial difference in relative volume and surface area for each of these compartment geometries (Fig. 5c). In the case where the total compartment surface area of MTs is the same

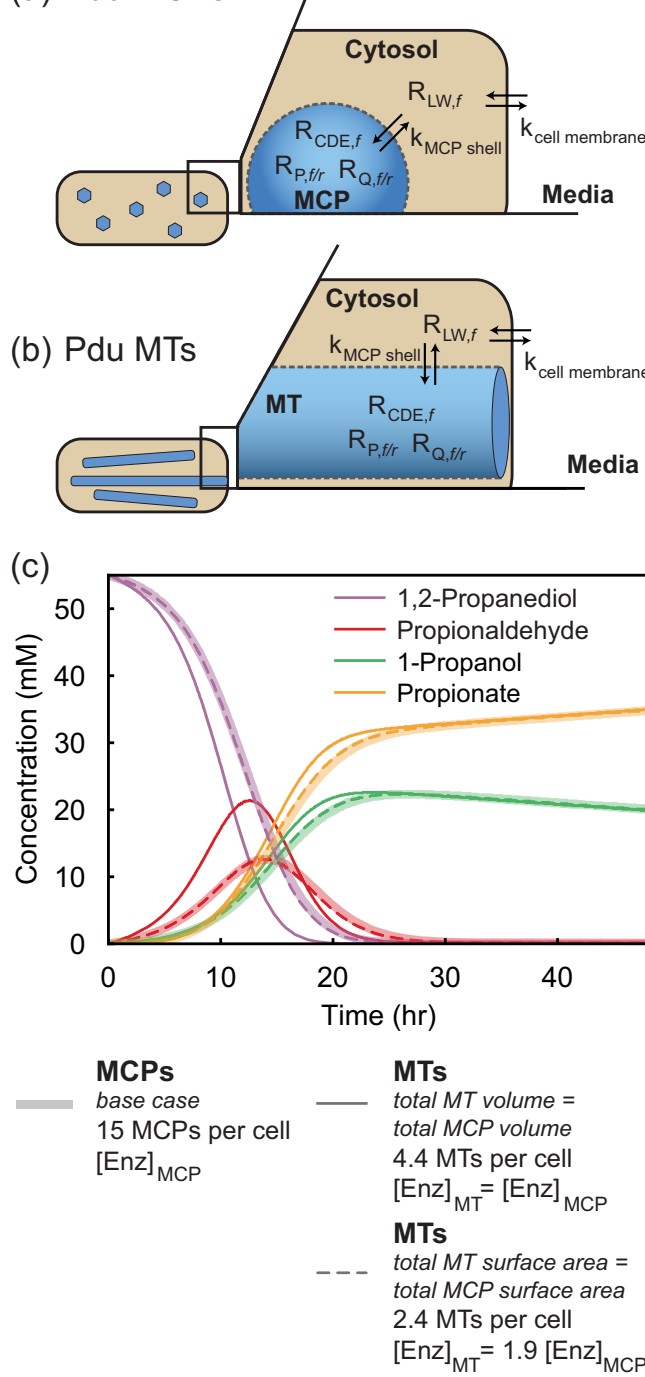

**Fig. 5 Systems-level kinetic model of the 1,2-propanediol pathway to explore the impact of compartment geometry on pathway performance. a** Spherical MCP model, where R refers to a reaction by a given enzyme, and k refers to diffusion across a barrier (the compartment shell or cell membrane) with a given permeability. **b** Cylindrical Pdu MT model. **c** Representative metabolite profiles in external media produced by the model for the spherical MCP base case (thick, slightly transparent lines), and limiting cylindrical MT cases (thinner, darker lines--solid for the case where internal MT volume is assumed the same as total MCP volume, and dashed for the case where total MT surface area is assumed to be the same as total MCP surface area). Data presented are concentration profiles calculated deterministically assuming an initial condition of 55 mM 1,2-propanediol in external media. Source data are provided as a Source Data file.

as spherical MCPs, but enzyme concentration increases 1.9-fold, we see little to no change in the metabolite profiles compared to the spherical MCP case (Fig. 5c). When total internal compartment volume is the same in MTs and MCPs, there is 1.9 times more surface area for metabolite diffusion in MTs. As a result of this increased available surface area for diffusion, 1,2-propanediol can more readily diffuse into the MT, where PduCDE rapidly converts it to propionaldehyde. This leads to increased peak propionaldehyde levels (Fig. 5c).

The experimental Pdu MT data show a slightly more rapid buildup of propionaldehyde in the ΔPduN strain than the wild type strain, suggesting that the overall surface area of these MTs may be slightly higher than MCPs. This conflicts with the 1,2-propanediol consumption data, in which the ΔPduN strain consumes 1,2-propanediol more slowly than the wild type strain, which is the opposite trend observed when surface area is increased in the model. Combined, this indicates that the geometry changes explored in the model cannot completely explain our experimental results; however, the modeling results do clearly describe how compartment geometry can impact encapsulated pathway kinetics. Specifically, these modeling results suggest that compartment surface area is a key parameter in dictating propionaldehyde buildup. Indeed, a local sensitivity analysis on all parameters used in the model reveals that total compartment surface area is the dominant morphological feature that controls external propionaldehyde buildup (see Supplementary Discussion 1 and Supplementary Fig. 4 and 5 for further details).

These results have both engineering and biological implications. In the engineering realm, they suggest that compartment shape and geometry offer a unique handle for tuning encapsulated pathway performance. Biologically, these results indicate that compartment geometry can play a role in the effectiveness of toxic intermediate retention, suggesting a rationale beyond strict closure for the formation of more spherical compartment structures.

**An elongated cell phenotype allows detection of pentamer incorporation**. Having established the importance of vertex protein PduN in MCP closure and the impact of compartment geometry on encapsulated pathway performance, we next developed a microscopy-based assay for screening Pdu MT versus Pdu MCP formation. This assay is based on the observation that Pdu MT formation in vivo results in an elongated, linked cell phenotype (Fig. 6a).

Analysis of microscopy data on cells expressing Pdu MCPs and Pdu MTs confirms that the elongated cell phenotype is specifically associated with formation of Pdu MTs. We performed microscopy and measured both the length and number of cells per chain of at least 100 cells over three biological replicates in both an MCP-forming strain (WT) and a MT-forming strain (ΔPduN). We find that the cell length is significantly different in MCP-forming (1.8 ± 0.4 μm, error is standard deviation) and MT-forming (8 ± 6 μm, error is standard deviation) strains (*p* < 0.0001) (Fig. 6b). Furthermore, more than 80% of the MT-expressing cells form chains of 3 or more cells, whereas cells expressing Pdu MCPs are always either single cells or double cells that are in the process of properly dividing (Fig. 6c). Thus, both cell length and percentage of linked cells provide a convenient readout for distinguishing between strains expressing Pdu MCPs and Pdu MTs. Notably, this linked cell phenotype is similar to that observed when self-assembling shell proteins PduA and PduJ are overexpressed in *E. coli*[23], a phenotype that has enabled rapid evaluation of the self-assembly propensity of point mutants of these hexamers[44]. We thus hypothesized that a similar strategy

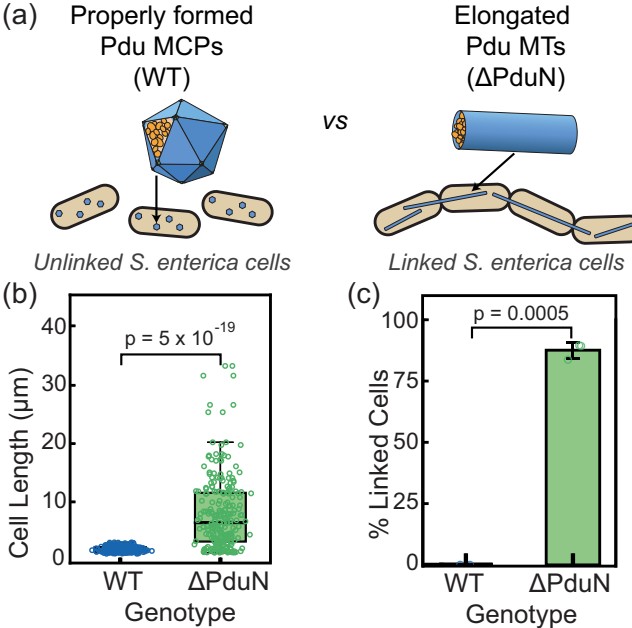

**Fig. 6 The formation of Pdu MTs leads to a linked cell phenotype that allows distinction between cells expressing Pdu MCPs versus Pdu MTs.** **a** Schematic of phenotypes associated with Pdu MCP and Pdu MT formation. **b** Box and whisker plot of length of cells expressing properly formed MCPs (WT) and cells expressing MTs (ΔPduN). Cell measurements were made over three biological replicates, with a total of 171 WT cells measured and 121 ΔPduN cells measured. The $p$ value displayed was calculated using a 2-tailed $t$ test assuming unequal variances. Box extends from the first quartile to the third quartile of the data, the line indicates the median, and whiskers extend from the box by 1.5 times the inter-quartile range. **c** Percentage of cells that contained 3 or more linkages (defined as cell bodies split by a clear cleavage furrow) in MCP-forming (WT) and MT-forming (ΔPduN) strains. Bars represent mean values of percentage linked cells over three biological replicates. Error bars represent standard deviation over three biological replicates. The $p$ value dispayed was calculated using a 2-tailed $t$ test assuming unequal variances. Source data for panels (**b**) and (**c**) are provided as a Source Data file.

could be leveraged to screen for Pdu MT formation. Importantly, we know that Pdu MCPs form when PduN incorporates into the compartment shell, and Pdu MTs form when PduN does not incorporate into the compartment shell. Consequently, we can use this MT-related phenotype to determine whether PduN point mutations prevent or permit incorporation into the compartment shell—linked cells would result from a non-incorporating PduN point mutant that causes MT formation, whereas unlinked cells would result from a PduN point mutant that correctly incorporates into MCPs.

We illustrate the utility of this phenotypic readout for screening PduN incorporation by assaying point mutant libraries of two residues in PduN—a glycine at position 52 in PduN (G52) and a threonine at position 88 (T88). We hypothesized that G52 would be highly immutable, as it is buried in the predicted interface between PduN and PduA (Fig. 7a), and the addition of any side chain group would be expected to sterically disrupt the stability of this interface[15]. As expected, most mutations at this residue result in a high population of linked cells (>60%, Fig. 7b, top), indicating that these point mutants are forming Pdu MTs. Indeed, fluorescence and electron microscopy confirm that cells expressing the *pdu* operon with PduN-G52C contain elongated Pdu MT structures (Fig. 7c). The prevalence of the linked cell phenotype, associated

with MT formation, in all PduN G52 point mutants suggests that these mutations do not permit incorporation of PduN into the MCP shell, and thus show that the G52 residue is highly immutable. Interestingly, one point mutant, G52N, in which the glycine is mutated to asparagine, shows a lower percentage of linked cells than the PduN knockout ($p < 0.01$, two-tailed $t$-test assuming unequal variances). Fluorescence microscopy on this point mutant suggests that there is a mixture of structures in these cells, evidenced by the combination of fluorescent puncta and streaks in these images (Fig. 7c). TEM on thin cell sections and purified compartments confirms this finding, showing the presence of both polyhedral and elongated structures (see Supplementary Discussion 2 for detailed discussion). This result suggests that the extent of cell elongation may be semi-quantitative, in that shorter, but still linked, cells contain a mixed population of Pdu MCPs and MTs.

Next, we investigated the mutability of residue T88 in PduN, which sits at the top of the PduA/PduN interface (Fig. 7a). We hypothesized that this residue would be amenable to mutation because it is not nestled in the pentamer/hexamer (PduN/PduA) interface. Cells expressing compartments with PduN T88 point mutants all resulted in low (<30%) linked cell populations (Fig. 7b, bottom). Even mutation to proline, an amino acid that typically disrupts protein structure, only results in 23 ± 1% of the cell population to be linked. This combination of results suggests that strains expressing compartments with PduN T88 point mutants generally produce well-formed Pdu MCPs. Thus, we conclude that this T88 residue in PduN is highly mutable. We confirmed the presence of well-formed Pdu MCPs in the ΔPduN::PduN-T88A strain using a combination of fluorescence microscopy, TEM, and SDS-PAGE (Fig. 7c). As expected, fluorescence microscopy shows punctate fluorescence distributed through the cell, an indicator of well-formed MCPs. TEM on thin cell sections confirms this finding, showing proteinaceous structures similar to those found in the wild type strain expressing Pdu MCPs (Figs. 7c, 2c). TEM on compartments purified from these strains confirm the formation of characteristic polyhedral structures (Fig. 7c), and Coomassie-stained SDS-PAGE on these same purified compartments shows that the protein content in these compartments is similar to that of compartments purified from cells containing a wild type *pdu* operon.

Combined, these results illustrate the utility of this elongated, linked cell phenotype in probing the molecular interactions important for compartment closure. We find that the assay readily distinguishes between PduN point mutants that are permissive and disruptive to compartment closure, evidenced by the differential trends observed in our G52 and T88 point mutant libraries. We expect that this microscopy-based assay for screening the functionality of PduN point mutants, tying PduN incorporation to a linked cell phenotype, will prove useful in both basic science and engineering contexts.

## Discussion

There is great interest in repurposing MCPs for metabolic engineering applications, where they have the potential to alleviate bottlenecks such as slow pathway kinetics, toxic intermediate buildup, and cofactor competition[2,11]. While strides have been made in loading non-native cargo into these systems in a controlled fashion[40,51–56], the selection criteria for an MCP system in any given engineering application is lacking. This includes MCP features such as size and morphology. Here, we report an in-depth characterization of an alternative Pdu compartment geometry, Pdu MTs, which form when vertex protein PduN cannot incorporate into the Pdu shell. Intriguingly, this shift in morphology upon loss of BMV-containing proteins is not universal

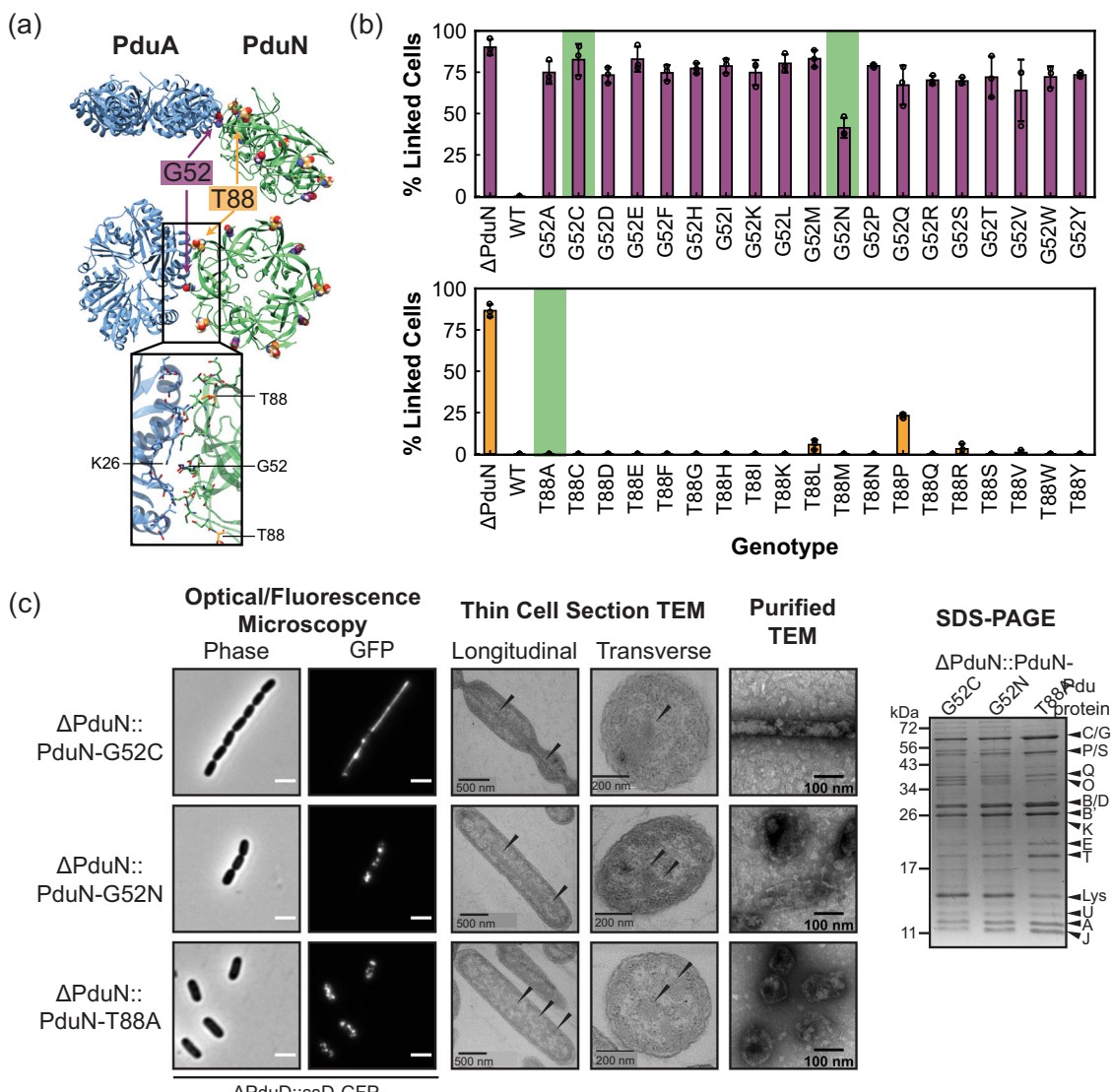

**Fig. 7 Interrogating the mutability of two PduN residues using the linked cell phenotype readout. a** Ribbon structure of the PduA-PduN interface (taken from the 40° bending angle simulations shown in Fig. 3), highlighting the two residues mutated in this work--glycine 52 (G52) in purple, which is buried in the interface, and threonine 88 (T88) in yellow, which sits at the top of the PduA-PduN interface. **b** Percent linked cell populations (defined as in Fig. 6) for strains expressing the *pdu* operon with all possible point mutations at position 52 in PduN (top) and position 88 in PduN (bottom). Bars represent mean percentage of linked cells over three biological replicates. Error bars indicate standard deviation in percent linked cells over three biological replicates. At least 100 cells were counted in total for each point mutant. The specific number of cells counted for each point mutant is available in Supplementary Table 2. Green highlights indicate point mutants that were selected for more detailed validation of Pdu MCP versus Pdu MT formation. **c** Detailed characterization of the Pdu compartment or tube structures formed in the presence of different PduN point mutants. Scale bars in optical and fluorescence micrographs are 2 μm. Arrows in thin cell section micrographs indicate protein-rich structures. Coomassie-stained SDS-PAGE gel is labeled with the bands expected for various Pdu proteins or lysozyme (Lys). Similar results were observed across three independent biological replicates for optical microscopy and SDS-PAGE experiments. TEM imaging of purified compartment structures and of thin cell sections was only conducted once. Source data are provided as a Source Data file.

across compartment systems—in the absence of vertex proteins, β-carboxysomes form elongated structures similar to Pdu MTs[34], but α-carboxysomes predominantly form regular icosahedra[35]. Further, other metabolosomes can form closed icosahedra in the absence of pentamers[53,54,57,58]. We hypothesize that this may be a consequence of the molecular interactions between shell proteins, specifically the preferred bending angle between these shell proteins. On this front, we anticipate that MD simulations can provide key insights towards understanding differences between compartment systems.

Comparison of growth and pathway performance in cells expressing Pdu MTs and Pdu MCPs showed that Pdu MTs

prevent buildup of the toxic propionaldehyde intermediate in the native Pdu pathway. In conjunction with our systems-level kinetic model, this result suggests that Pdu MTs provide a diffusive barrier between the cytosol and the encapsulated enzyme core. However, we note that the morphology change from spherical MCPs to cylindrical MTs necessarily changes the surface area to volume ratio of the compartment. We expect that the different surface area to volume ratio made available by these Pdu MT structures will prove beneficial to engineered encapsulated pathways with different kinetic profiles. Future analysis of different encapsulated pathways across different compartment geometries will provide valuable insight in this regard.

The discovery of a genotype-phenotype link for Pdu MT formation provides a microscopy-based method for screening formation of these structures that does not require lengthy purification nor specialized equipment. In the context of understanding MCP assembly, this enables rapid characterization of the fitness and mutability of different interacting residues in shell proteins like PduN and PduA. For engineering purposes, this assay provides a facile method for screening whether point mutations to PduN allow PduN incorporation into the Pdu MCP shell. This could prove useful for incorporating reactive handles into this shell protein, which is particularly promising in light of the potential utility of non-canonical amino acid incorporation into MCPs[59,60]. Combined, this brings us one step closer to realizing the full potential of MCPs as engineerable bionanoreactors.

## Methods

**Plasmid creation.** The sequence encoding for PduN was cloned into a Golden Gate-compatible pBAD33t parent vector (p15a origin of replication, chloramphenicol resistance selection cassette, arabinose-inducible promoter) using Golden Gate cloning[61]. A C-terminal 6xHistidine or FLAG tag was added to the PduN sequence during cloning. All primers are listed in Supplementary Data 3, and all plasmids generated are listed in Supplementary Data 2. All cloning was performed using *Escherichia coli* DH10b cells.

All PduN point mutants were first generated in the PduN sequence cloned into the aforementioned pBAD33t vector prior to integration into the *Salmonella* genome (see below for recombineering methods). PduN point mutants at glycine 52 (G52) were generated using QuikChange site-directed mutagenesis with KOD Hot Start DNA polymerase (Sigma–Aldrich) on a PduN sequence. PduN point mutants at threonine 88 (T88) were generated using the entry vector method described previously[62,63]. Briefly, Gibson assembly was used to replace amino acids 80-91 in PduN with a constitutively active GFP gene flanked by two BsaI sites. Point mutants were then ordered as single-stranded DNA primers flanked by BsaI cut sites complementary to those in the entry vector. The reverse strand was filled in using PCR with 12-mers directed to the Golden Gate cut sites. Double stranded DNA was purified using a PCR cleanup kit and used in Golden Gate assembly reactions with the entry vector. Candidate clones were screened by green-white screening. The sequences of all point mutant plasmids generated were confirmed by Sanger sequencing (Genewiz).

**Strain generation.** All strains used in this work are listed in Supplementary Data 1. Recombineering was performed using λ red recombineering as previously described[64]. Briefly, genetic modifications were made by first replacing the gene at the locus of interest with a cassette encoding for a chloramphenicol resistant gene (*cat*) and sucrose sensitive gene (*sacB*). This *cat/sacB* selection cassette was amplified from the *TUC01* genome[64,65] using primers that added homology to the genomic locus of interest. Genomic incorporation of this cassette was confirmed by selection on lysogeny broth (LB)-Agar supplemented with 10 μg/mL chloramphenicol, followed by sucrose sensitivity screens on select colonies on LB-Agar plates supplemented with 6% (w/w) sucrose. Next, the *cat/sacB* selection cassette was replaced with the DNA encoding for the desired gene, using either single-stranded DNA for knockouts, or purified PCR products for full genes. PCR products used for PduN point mutant incorporation at the *pduN* locus or ssD-GFP incorporation at the *pduD* locus were amplified from the respective plasmid using primers that added homology to the genomic locus of interest. Replacement of the *cat/sacB* selection cassette with the DNA of interest was selected for using sucrose sensitivity. Clones were sequence confirmed by Sanger sequencing on PCR products amplified from the genome at the locus of interest.

**Cell growth.** *S. enterica* growths for all non-growth curve experiments were conducted as follows. Overnight cultures were inoculated from single colonies into 5 mL of LB, Lennox formulation (LB-L) and grown at 30 °C, 225 rpm for 24 h. For experiments in which PduN-FLAG was expressed off a plasmid, overnight cultures were supplemented with 34 μg/mL chloramphenicol. Overnights were subcultured 1:1000 into 5 mL No Carbon Essential (NCE) media (29 mM potassium phosphate monobasic, 34 mM potassium phosphate dibasic, 17 mM sodium ammonium hydrogen phosphate) supplemented with 50 μM ferric citrate, 1 mM magnesium sulfate, 42 mM succinate as a carbon source, and 55 mM 1,2-propanediol as an inducer of the *pdu* operon. For experiments in which PduN-FLAG was supplemented off a plasmid, media was supplemented with 34 μg/mL chloramphenicol and the specified amount of arabinose, both added at the time of subculture. Cells were grown in a 24-well block for 16 h at 37 °C, 225 rpm and then prepared for imaging experiments (see fluorescence/phase microscopy and thin cell section TEM below).

**Compartment expression and purification.** MCP expression and purification was performed using differential centrifugation as previously described[39,66]. Briefly, overnight cultures were started from a single colony in 5 mL LB-L and grown at 37 °C, 225 rpm. Overnights were supplemented with 34 μg/mL chloramphenicol for strains in which PduN-FLAG was expressed off a plasmid. Overnights were subcultured 1:1000 into 200 mL of NCE (again, supplemented with ferric citrate, magnesium sulfate, succinate, and 1,2-propanediol, as above) and grown at 37 °C, 225 rpm until $OD_{600}$ reached 1–1.5. Cultures for strains in which PduN-FLAG was expressed off a plasmid were supplemented with 34 μg/mL chloramphenicol and 0.02% (w/w) arabinose, both added at the time of subculture. Cells were harvested by centrifugation (4500 × $g$, 5 min), and resuspended in lysis buffer (32 mM Tris-HCl, 200 mM potassium chloride (KCl), 5 mM magnesium chloride (MgCl₂), 0.6% (v/v) 1,2-propanediol, 0.6% (w/w) octylthioglucoside (OTG), 5 mM β-mercaptoethanol, 0.8 mg/mL lysozyme (Thermo Fisher Scientific), 0.04 units/mL DNase I (New England Biolabs, Inc.) pH 7.5–8.0). Resuspended cells were allowed to lyse in this buffer by incubating at room temperature for 30 min. Lysate was then clarified by two rounds of centrifugation (12,000 × $g$, 5 min, 4 °C). MCPs or MTs were separated from lysate by ultracentrifugation in a swinging bucket rotor (21,000 × $g$, 20 min, 4 °C), washed with buffer (32 mM Tris-HCl, 200 mM KCl, 5 mM MgCl₂, 0.6% (v/v) 1,2-propanediol, 0.6% (w/w) OTG, pH 7.5–8.0), and then centrifuged again (21,000 × $g$, 20 min, 4 °C). MCP or MT pellets were then resuspended in buffer (50 mM Tris-HCl, 50 mM KCl, 5 mM MgCl₂, 1% (v/v) 1,2-propanediol, pH 8.0). Remaining cell debris in the sample was then removed by three 1 min centrifugations at 12,000 × $g$. Purified MCPs or MTs were stored at 4 °C until use.

**Gel electrophoresis and western blotting.** Protein content of purified MCP or MT samples was assessed using sodium dodecyl sulfate-polyacrylamide gel electrophoresis (SDS-PAGE) paired with Coomassie-blue staining or anti-FLAG western blotting. Loading of purified MCP or MT samples was normalized by total protein concentration, as determined by bicinchoninic acid assay, such that 1.5 μg protein was loaded per lane. Samples were diluted into Laemmli buffer and heated at 95 °C for 5 min prior to loading. Samples were separated on 15% (w/w) polyacrylamide Tris-glycine mini gels for 90 min at 120 V. Protein was visualized by staining with Coomassie Brilliant Blue R-250.

Samples for western blot were prepared and separated by SDS-PAGE as above. Samples were transferred to a polyvinylidene fluoride (PVDF) membrane using a Bio-Rad Transblot SD at 25 V, 150 mA, for 35 min. The membrane was blocked in TBS-T (20 mM Tris, 150 mM sodium chloride (NaCl), 0.05% (v/v) Tween 20, pH 7.5)) with 5% (w/w) dry milk for 1 h at room temperature. The membrane was then probed with a mouse anti-FLAG primary antibody (MilliporeSigma Cat# F3165) diluted 1:6666 in TBS-T with 1% (w/w) dry milk for 1 h at room temperature. The membrane was washed with TBS-T and then incubated for 30 min at room temperature with a goat anti-mouse-horseradish peroxidase secondary antibody (Invitrogen Cat# 32430) diluted 1:1000 in TBS-T. Finally, the membrane was washed with TBS-T and subsequently developed using SuperSignal™ West Pico PLUS Chemiluminescent Substrate (Thermo Fisher Scientific) and imaged using a Bio-Rad ChemiDoc XRS + System equipped with Bio-Rad Image Lab software.

**Phase contrast and fluorescence microscopy.** Cells were prepared for microscopy on Fisherbrand™ frosted microscope slides (Thermo Fisher Scientific Cat# 12-550-343), and sandwiched between the slide and a 22 × 22 mm, #1.5 thickness coverslip (VWR Cat# 16004-302). Coverslips and microscope slides were cleaned with ethanol before use. Cells were imaged on a Nikon Eclipse Ni-U upright microscope using an 100X oil immersion objective using an Andor Clara digital camera. NIS Elements Software (Nikon) was used for image acquisition. GFP fluorescence images were acquired with a C-FL Endow GFP HYQ bandpass filter. A 200 ms exposure time was used for all fluorescence images. Brightness and contrast in images across a given experiment were adjusted to the same values in ImageJ[67]. Cell length was quantified using the segmented line tool in ImageJ[67]. Identification of linked cells was done as previously described[23]. Briefly, any cell body containing three or more cell bodies divided by an identifiable cleavage furrow were counted as linked, whereas cell bodies containing two or fewer cells (or one or fewer cleavage furrows) were considered unlinked. Each cell body in a set of linked cells was counted as a single cell event.

**Transmission electron microscopy.** Purified MT or MCP samples were prepared and imaged using negative-stain TEM as previously described[20]. Briefly, samples were fixed on 400 mesh Formvar-coated copper grids (EMS Cat# FF400-Cu) using a 2% (v/v) glutaraldehyde in water solution. Fixed samples were washed with MilliQ™ pure water and stained with 1% (w/w) uranyl acetate solution. Grids were dried and stored prior to imaging. Grids were imaged using a JEOL 1230 transmission electron microscope equipped with a Gatan 831 bottom-mounted CCD camera. Measurements of MT diameter were performed using the segmented line tool in ImageJ[67].

Sample cells were fixed in 2% (v/v) Paraformaldehyde and 2.5% (v/v) EM Grade Glutaraldehyde in a 0.1 M PIPES buffer, pH 7.4 and chemically processed with Osmium Tetroxide, dehydrated with ethanol and acetone, and infiltrated with EMBed812 epoxy resin within an mPrep ASP1000 Automated Sample Processor. The infiltrated samples were embedded in block molds with pure resin and

polymerized at 60 °C for 48 h. Ultrathin 60 nm sections of the embedded cells were cut at using a Leica UC7 Ultramicrotome and a DiATOME 35 degree diamond knife. Sections were collected on Cu slotted grids with a formvar/carbon membrane and stained with Uranyl Acetate and Lead Citrate to enhance inherent contrast within the electron microscope. Sample sections were loaded into a Hitachi HD2300 cFEG STEM at 200 kV and imaged with the TE phase contrast and HAADF Z-contrast detectors. Image data was collected with DigiScan, an e- beam rastering data collection system within Gatan Digital Micrograph.

**Growth assay and metabolite quantification**. Growth assays on 1,2-propanediol as a sole carbon source were performed as previously described[23,49]. Briefly, overnights in 5 mL terrific broth (TB) without glycerol (Dot Scientific, Inc.) were inoculated from single colonies and grown for 15–16 h at 37 °C with orbital shaking at 225 rpm. Overnights were subcultured to an $OD_{600}$ of 0.05 into NCE media supplemented with 50 μM ferric citrate, 1 mM magnesium sulfate, 150 nM adenosylcobalamin, and 55 mM 1,2-propanediol. Cultures were grown in foil-capped 250 mL unbaffled Erlenmeyer flasks at 37 °C, 225 rpm.

At each time point, a 500 μL aliquot of culture was removed to quantify $OD_{600}$ and metabolite levels. $OD_{600}$ was measured on a BioTek Synergy HTX multi-mode plate reader and converted to equivalent $OD_{600}$ for a 1 cm pathlength. After nine hours, culture samples from all strains except ΔPocR were diluted 1:5 in fresh NCE prior to $OD_{600}$ measurement to ensure measurements were performed in the linear range of the instrument. Error bars on growth curves represent standard deviation over three biological replicates.

Cell culture sample not used for $OD_{600}$ measurement was centrifuged at $13,000 \times g$ for 5 min to remove cells. Supernatant was collected and frozen at −20 °C. Prior to HPLC analysis, samples were thawed and filtered (Corning™ Costar™ Spin-X LC filters). Filtered samples were analyzed on an Agilent 1260 HPLC system, separated using a Rezex™ ROA-Organic Acid H + (8%) LC Column (Phenomenex) at 35 °C. The separation was isocratic, with 5 mM sulfuric acid as the mobile phase, flowing at 0.4 mL/min. Metabolites were detected using a refractive index detector (RID)[19], where peaks in the RID spectrum were observed at 30 min for 1,2-propanediol, 33 min for propionate, 39 min for propionaldehyde, and 46 min for 1-propanol. Metabolite concentrations were calculated from peak areas calculated in Agilent ChemLab software using standards of the metabolites of interest (1,2-propanediol, propionaldehyde, propionate, 1-propanol) at 200, 100, 50, 20, and 5 mM. Error bars on reported metabolite concentrations represent the standard deviation over three biological replicates.

**All-atom molecular dynamics simulations**. The initial structure for the atomistic model of the PduA/PduN interface was generated as follows. The PduA structure was taken from PDB 3NGK[25]. The structure of the PduN subunit was estimated using the Phyre2 web portal[68]. The pentamer structure was then generated by aligning five copies of this PduN subunit structure to the BMC-P structure extracted from PDB 5V74[15] using the MatchMaker tool in UCSF Chimera[69,70]. This structure was then minimized using default parameters in UCSF Chimera's Minimize Structure tool. To build the PduA/PduN interface, a BMC-H/BMC-P interface was extracted from PDB 5V74, which is a solved crystal structure of a full microcompartment from *Haliangium ochraceum*[15]. Chimera's MatchMaker tool was then used to align the PduA hexamer and PduN pentamer to the BMC-H and BMC-P structures, respectively. The PduA/PduN interface structure was then minimized again using the default parameters in Chimera's Minimize Structure tool[69]. The PduA/PduA interface was generated in the same way, except using a BMC-H/BMC-H interface from the PDB 5V74 structure.

Prior to running simulations, the PduA/PduA and PduA/PduN models were solvated in water containing 100 mM NaCl. Using the GROMACS molecular dynamics engine[71], the system was subject to a 100 ps constant pressure, temperature (NPT) equilibration with the protein backbones restrained. Steered MD simulations were then run to create configurations where the proteins adopt many different bending angles or distances (depending on the nature of the calculation). For the calculation of the bending potential, the protein was constrained to move in only two dimensions such that only the bending angle between the two proteins could change. Umbrella sampling was then performed in the z-direction and mapped back onto the $\theta_B$ direction, converting the forces in the z-direction to those in the $\theta_B$ direction as described in Supplementary Method 1. For the total interaction strength of PduA/PduN standard umbrella sampling is applied. More details for AA MD simulations are provided in the Supplementary Method 1.

**Kinetic pathway modeling**. The kinetic model used to simulate the Pdu pathway was modified from previous work[36], and is described in full detail in Supplementary Method 2. Here, we moved beyond steady state analysis, and analyzed the metabolite profiles over time to mirror the growth conditions in our experiments. We also explicitly modeled the external media and accounted for increase in cell number based on the experimental growth curve data. We calculate the dynamics of metabolites 1,2-propanediol, propionaldehyde, propionyl-CoA, 1-propanol, and propionate within the MCP/MT interior, cytosol, and media. Noting that metabolite diffusion with each region is much faster than the enzyme kinetics and transport across the cell membrane or MT/MCP shell, we assume that each volume

is well-mixed and has no spatial variation. Therefore, the main consequence of the chosen geometry is the surface area and volume of each region.

Our model assumes that MCPs are spheres with a 140 nm diameter and MTs are cylinders with a 50 nm diameter and a length equal to that of the cell. Metabolites passively diffuse into spherical MCPs over the entire spherical surface area at a rate determined by the permeability of the shell. In MTs, metabolites can only diffuse into the cylindrical volume along the long axis of the cylinder, and not at the ends, again at a rate determined by the permeability of the shell. An alternative model considering diffusion out of the MT ends is described in detail in Supplementary Method 2. The permeability of the MCPs/MTs is assumed the same for all metabolites. A set number of MCPs or MTs exist in a cell at all times, set by the MCPs/MTs per cell parameter. Cells are assumed to be capsule shaped.

All enzymes are assumed to exhibit Michaelis Menten kinetics. We assume that the reactions catalyzed by PduCDE, PduP, and PduQ can only occur inside the MCP/MT, and that the reactions catalyzed by PduL and PduW that convert propionyl-CoA to propionate happen in a single step in the cytosol of the cell. The conversion of 1,2-propanediol to propionaldehyde by PduCDE is assumed irreversible. The conversions of propionaldehyde to either propionyl-CoA or 1-propanol by PduP and PduQ, respectively, are both considered reversible. The conversion of propionyl-CoA to propionate by PduL/PduW is assumed irreversible.

This model was implemented in Python[72], and is available on GitHub [https://github.com/cemills/MCP-vs-MT][73]. A detailed description of the equations is in Supplementary Method 2. The parameters are summarized in Supplementary Table 1.

**Reporting summary**. Further information on research design is available in the Nature Research Reporting Summary linked to this article.

## Data availability

All strains and plasmids used in this study are available upon request. We used PDBs 5V74[15] and 3NGK[25] in this work to generate starting coordinates for all-atom simulations performed in this work, as described in the Methods section. Source Data are provided with this paper.

## Code availability

Codes used for kinetic modeling are available on GitHub [https://github.com/cemills/MCP-vs-MT][73].

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

## Acknowledgements

The authors thank and acknowledge members of the Mangan, Olvera de la Cruz, and Tullman-Ercek groups for insightful discussions around this work. The authors specifically acknowledge Dr. Chris Jakobson for plasmids used in this study. This work was funded in part by the Army Research Office (grant W911NF-19-1-0298 to D.T.E. and M.C.J.) and the Department of Energy (grant DE-SC0019337 to D.T.E. and N.M.M. and grant DE-FG02-08ER46539 to M.O.d.l.C). N.W.K. was funded by the National Science Foundation Graduate Research Fellowship Program (grant DGE-1842165), and by the National Institutes of Health Training Grant (T32GM008449) via the Northwestern University Biotechnology Training Program. M.O.d.l.C. thanks the computational support of the Sherman Fairchild Foundation. This work made use of the BioCryo facility of Northwestern University's NUANCE Center, which has received support from the SHyNE Resource (NSF ECCS-2025633), the IIN, and Northwestern's MRSEC program (NSF DMR-1720139). Molecular graphics and analyses performed with UCSF Chimera, developed by the Resource for Biocomputing, Visualization, and Informatics at the University of California, San Francisco, with support from NIH P41-GM103311.

## Author contributions

C.E.M., C.W., N.W.K, A.D.J., M.O.d.l.C., and D.T.E. conceived this project. C.E.M., N.W.K., C.H.A., A.D.J., and E.W.R. performed experiments. C.W. performed atomistic simulations. All authors contributed to analysis and interpretation of data. A.G.A., S.S., and N.M.M. contributed to development of software used in the systems-level kinetic model. C.E.M. and D.T.E. wrote the manuscript. C.E.M., C.W., A.G.A., N.W.K, C.H.A., A.D.J., E.W.R., S.S., M.C.J., N.M.M., M.O.d.l.C., and D.T.E. reviewed and contributed to the manuscript.

## Competing interests

The authors declare no competing interests.
