## [Peer Review File · Nature Communications]

Vertex protein PduN tunes encapsulated pathway performance by dictating bacterial metabolosome morphologyReviewers' Comments:

Reviewer #1:

Remarks to the Author:

This manuscript explores the consequences of altering the morphology of the Pdu bacterial microcompartment. The change is achieved by removing the PduN protein. The authors rely on the speculation that PduN forms pentamers---I am not aware of any direct evidence that this is the case--to hypothesize that the microcompartment structure resulting from PduN deletion will differ from the characteristic icosahedral structure that forms when the full operon is expressed. There is strong experimental evidence throughout characterizing the resulting structures, which form extended cylindrical tubes visible in TEM data.

In addition to the experimental characterization of Δ PduN tubule properties and the consequences for cell metabolism, the authors conduct two computational investigations: one that probes the molecular interactions between cell components and another that investigates the consequences of the geometry for the metabolism of 1,2-propanediol. My review will focus primarily on issues relating to the molecular dynamics and systems-level model.

The authors use steered molecular dynamics to estimate the potential of mean force of both the PduN-PduA and the PduA-PduA binding angle. As the authors point out, the difference between the 0 and 34 degree configurations is ~ 2 kT with 1 kT error bars, so this calculation suggests that, at least within the accuracy of the model, these configurations interconvert frequently. The authors make no comment on the fact, however, that the lowest free energy configuration according to their data has a binding angle of roughly 75 degrees, which is biologically implausible but should nevertheless be addressed in the text. The PduN model binds to PduA with a much better-defined geometry. What is the angle that should be expected for an idealized icosahedral geometry? How much would the other coordinating configurations (other surrounding hexamers) be expected to affect this calculation?

The experimental characterization of the growth rate and metabolism shows that, while unbroken MCPs are important for growth and performance, Δ PduN has modest effects. A systems-level kinetic model is used to study the impact of the geometry of the microcompartment on the metabolic activity of the mutant cells. The model itself has been described in detail elsewhere; it makes reasonable assumptions about the kinetics of the reactions and assumes passive transport through perfectly closed shells. Experimentally, somewhat subtle changes in 1,2-propanediol are observed with the MT geometry (slightly slower consumption). In the systems model, this trend is reversed in the case of total volume and not obviously different from wild type when surface area is equated. So, while the geometry does indeed affect the results, it does so in a way that is consistent with the experimental observation of propionaldehyde production but inconsistent with 1,2-propanediol consumption. I think this point should be clearly addressed in the main text; currently there is no discussion of this point in the main text nor the supplement. I find the explanation of the eventual consumption of the propionate and 1-propanol to be adequate. The fact, on the other hand, that propionaldehyde concentration is off by an order of magnitude requires a more serious discussion. If the subtle changes described in the supplement about temperature and shaking can affect concentration by that amount, the authors should make this clearer. The last point about the calculation that I do not quite understand: the cylindrical MT structures cannot be fully enclosed without pentamers. This means that the observed structure should be expected to be more like a straw than an enclosed cylinder. If diffusion of small molecules within the MCP is rapid, how does this structure provide the necessary passive barrier?

On the whole, while I am convinced that the authors have found a robust method for altering the geometry of the Pdu MCPs, the metabolic consequences of this change appear minor (one data point of 1,2-propanediol and propionaldehyde concentration differ outside error bars in Fig. 3 c). Rather than provide a clear mechanistic explanation of these small changes, the systems-level model indicates that a purely geometric explanation leads to changes that are qualitatively distinct from

those in the experiment. The paradigm of modulating geometry of subcellular structures with the goal of impacting behavior, however, is an important, timely, and significant one.

Reviewer #2:

Remarks to the Author:

The authors report a comprehensive study on the interplay between structure and function of microbes microcompartments (MCPs) and how this knowledge can be used to characterize and leverage this sub-cellular organization. The study centers on pdu MCP as a model system, and specifically characterizes the role of two protein aggregates in the overall morphology of the compartment. PduN has a pentameric structure and it is known to form the vertices in encapsulated MCPs in *Salmonella enterica*. Whereas PduA has a hexameric spatial arrangement, and was used to model the majority of the "planar" constituents of the MCP at hand. The manuscript presents a systematic approach to study the molecular level details of PduN on the morphology of the compartment using experimental and computational approaches to differentiate encapsulated MCPs vs microtubules. The amount of mutants considered for the study is impressive and attests to the determination of the team to characterize the molecular factors that modulate microcompartment morphology and "composition."

The manuscript as a whole is very easy to follow without taking away the technical aspects and scientific value of the study. The results are presented for the model system as well as the potential applications and broader impact in nanomaterials design, complex macromolecular assemblies (such as viral capsids), and modulation of microcompartment encapsulation metabolic profile/signature. One of the main outcomes presented by the authors is the use of specific techniques in (fluorescence) microscopy to track and characterize protein oligomers in the different encapsulated morphologies. This study is of interest to the wider scientific community and I recommend it for publication with minor reviews listed below.

1. The methods are all thoroughly described in the main manuscript and supporting documents. However, there are details missing about the computational approach; in line 686 the authors state more details about the MD simulations are provided in the supplement, but there are no such details in either Sup Doc 1 or 2. The authors should provide the following information to ensure reproducibility:

- a. Dynamic ensemble and force field; any other pertinent settings to run the simulation (integration algorithms)
- b. Production simulation time (beyond initial relaxation of 100ps)
- c. Number of windows used to compute the PMF and details for umbrella sampling
- d. Assess/discuss convergence of the PMFs presented in figure 3.
- e. Were any replicas run for the trajectories?

2. The section labeled "discussion" should be Conclusions or Outlook. The authors provide excellent context and evidence for their hypotheses and conclusions in the Results section, which in fact can perfectly be labeled as "Results and discussion."

Reviewer #3:

Remarks to the Author:

The authors present a very clear manuscript describing the impact of changes to the PduN vertex protein on the morphology of the *Salmonella enteric* Pdu micro compartment.

Elucidating the structure and function of bacterial metabolosomes is of fundamental importance in understanding the role of these in the metabolism of niche carbon sources, particularly in enteric

pathogens like *Salmonella enterica*. Further to this basic research, metabolosomes present biotechnological important 'reaction' vessels for metabolic engineering.

The results presented by the authors demonstrate the role of the PduN vertex protein in determining the closure of an icosahedral metabolosome, and in its absence the formation of tubular structures.

This manuscript is very well written and presents important results and new methodological approaches for screening variant metabolosomes microscopically. I would recommend publication with only minor changes for clarity.

In terms of clarifications/additions to enhance the manuscript I have the following minor comments to the authors.

Figure 2 - panel b - note the addition of lysozyme in the figure legend (I am assuming the band indicated 'Lys' is lysozyme. In panels C/D it is very hard to identify the GFP fluorescence indicated in the micrographs. Can this figure be adjusted in some way to enhance the contrast/signal from these micrographs?

Figure 4: Panel b - can the specific growth rates for the various mutants be calculated to give solid values to the differences in growth for the mutants? From a purely visual estimation the exponential growth rates are very similar, but it would be good to have a number to compare these.

Figure 5 - the assumption that there is no significant diffusion of substrates across the MT ends is not clear from my reading of the manuscript, or the supplementary material. Could you be more explicit in your reasoning for this assumption. Do you suggest that the MT ends have no shell proteins, or are they capped in some way? Either way one would imagine that there would be some diffusion across this area.

Figure 7 - GFP panels do not have sufficient contrast.

Line 489 - repetition of 'towards a fundamental understanding' leads to a lack of clarity of your message. Can you reword for clarity?

Methods - line 524 - can you give a reference to the TUC01.

Reviewer #4:

Remarks to the Author:

This manuscript presents a genetic, metabolomics, and computational modeling analysis of the role played by the pentameric shell protein PduN in the pdu microcompartment. Experimental data show that deletion of the pentamer tends to give elongated tubular structures, sometimes interfering with complete cell division. Despite the structural defects, the aberrant containers appear to offer reasonably good metabolic performance compared to complete absence of microcompartments, though with some more nuanced differences compared to wild-type in terms of product balance, which could be directly related to biochemical defects or to secondary mechanisms. The study is supported by diffusion and kinetic modeling. Similarly, the findings there show some dependence on shape and size parameters rather than major deviations compared to containers of wild-type shape. A weakness of the modeling approach is that it appears the openness of the tubes could not be modeled effectively with the approach taken and it is hard to know if diffusion through open ends would have had an important effect in a computational approach where that could be accounted for. There is also a molecular dynamics analysis of the energetics of association between a shell protein hexamer and a pentamer, focusing on bending energetics. This part of the study, which is a bit tangential to the main gene function and metabolic aspects of the paper, appears to have a problem with the mathematical

treatment of energetics vs bending angle, which the authors will need to revisit. Overall, the work brings a combination of sophisticated tools to bear on some open questions about function in bacterial microcompartments. The findings are not too surprising, but sensible and worth publishing in a good venue. Much of the value of the study could be in its approaches, including computational methods, which might find use in other complex systems. With that in mind, I think the work could have positive impact if technical issues (below) can be ironed out.

Major point:

There is some complexity in how the authors have sought to analyze the energetics of hexamer/pentamer bending on the basis of MD simulations carried out over a series of starting conformations at a range of bending angles. The description of how those calculations were analyzed is tough to follow, but after a few passes I was able to come (I think) to an understanding. Based on that, my conclusion is that an important mathematical error was made in going from the analysis of local MD results (as a function of z) to what is supposed to amount to a kind of path integral over the bending angle θ . The key equation, around line 15-16 in the supplement, would work if the energy V was dependent on z and not any other (rigid body) variable. But that is not the case. In order to get a correct treatment of V by integrating approximate values of dV as θ changes, the authors need to start with the complete differential of V with respect to the rigid body positions. If it's reasonable to treat that as a z component and a radial component (in the y direction, as the authors have set things up), then the starting point would be

$$dV/d(\theta) = dV/dz dz/d(\theta) + dV/dy dy/d(\theta)$$

In their treatment, the authors have left out the second term. I believe that after thinking it through the authors will agree that it is not possible to obtain a valid measure of the difference in V between an initial and final value of θ by only considering the partial derivative of V with respect to z , but not y . It is easy to construct a trivial scenario to show this. E.g. if a function V varies as a function of y but is independent of z , which though not likely is entirely permissible mathematically, then the authors' equation (using only the first term), would predict no change in V from the beginning to ending values of θ , even though y (and hence V) would have changed. If the authors are to rescue their attempt to obtain V as a function of θ , then they would need to reexamine their MD data to extract estimates of dV/dy . Perhaps the numerical effects of including the dependence on y will be modest. It is hard to be sure. But the treatment does require that term. I spent quite a bit of time coming to grips with the treatment, so I am 99% confident of the conclusion here, unless somehow the authors can show what happens to the missing term in the total derivative. [There doesn't seem to be any reason to think it would be much smaller than the term that was included.]

Minor comment:

In the same equation in the supplement: in the end I suppose it is cosmetic, but the equation as written seems to introduce some unit problems in the intermediate terms (which then cancel). As written, $dV/d(\theta)$ would not agree with the units for a Newtonian force. It seems that a dependence on radius should probably be introduced formally.

F should probably be written as $= -(1/r) dV/d(\theta)$. And in the next step, $dz/d(\theta)$ should be $r \cdot \cos(\theta)$. The $(1/r)$ and r then cancel.

[As noted above, it seems a correct treatment would require terms for dependence on y , treated similarly.]

We note that all line numbers in this response refer to the marked up document ("Mills_PduN_Resubmission_TrackedChanges.pdf")

Reviewer #1 (Remarks to the Author):

This manuscript explores the consequences of altering the morphology of the Pdu bacterial microcompartment. The change is achieved by removing the PduN protein. The authors rely on the speculation that PduN forms pentamers---I am not aware of any direct evidence that this is the case---to hypothesize that the microcompartment structure resulting from PduN deletion will differ from the characteristic icosahedral structure that forms when the full operon is expressed. There is strong experimental evidence throughout characterizing the resulting structures, which form extended cylindrical tubes visible in TEM data.

*We agree with the reviewer that crystal structure and multimerization data specifically indicating that PduN is a pentamer has yet to be published; however, we feel that acknowledgement of this uncertainty is captured in our introduction of the PduN protein in the paper (lines 71-72), where we state that that "pduN is the sole bacterial microcompartment vertex (BMV) pfam03319 gene in the pdu operon and is thus **expected** to form pentamers that cap the vertices of the Pdu MCP."*

However, given:

*(1) the extensive crystal structure data on other BMV-containing proteins¹⁻⁶, including full microcompartment structures that show these BMV-containing proteins capping microcompartment vertices as a direct consequence of their pentameric form^{2,7}, and
(2) the fact that multiple published bioinformatics studies identifying microcompartment systems delineate functional microcompartment operons as those containing genes encoding for at least one BMC-containing and one BMV-containing domain based on the assumption that BMV-containing proteins form the pentamers necessary for vertex formation in compartments, we do not agree that the assumption that PduN forms pentamers is speculative.*

We have added further references to our claim that we expect this to be true to further support this point.

In addition to the experimental characterization of Δ PduN tubule properties and the consequences for cell metabolism, the authors conduct two computational investigations: one that probes the molecular interactions between cell components and another that investigates the consequences of the geometry for the metabolism of 1,2-propanediol. My review will focus primarily on issues relating to the molecular dynamics and systems-level model.

The authors use steered molecular dynamics to estimate the potential of mean force of both the PduN-PduA and the PduA-PduA binding angle. As the authors point out, the difference between the 0 and 34 degree configurations is ~2 kT with 1 kT error bars, so this calculation suggests that, at least within the accuracy of the model, these configurations interconvert frequently. The authors make no comment on the fact, however, that the lowest free energy configuration according to their data has a binding angle of roughly 75 degrees, which is biologically implausible but should nevertheless be addressed in the text.

We thank reviewers for noting this feature of the data. We have added discussion to lines 202-208 to this point:

“Second, we note that a second minimum exists at a PduA/PduA interaction angle of $\sim 70^\circ$; however, given that this bending angle would not permit assembly of larger icosahedra or polyhedra like those formed in the Pdu MCP system, we do not believe it is physically relevant to the discussion here.”

The PduN model binds to PduA with a much better-defined geometry. What is the angle that should be expected for an idealized icosahedral geometry? How much would the other coordinating configurations (other surrounding hexamers) be expected to affect this calculation?

*We thank the reviewer for this insightful question. To the first point, a crystal structure of the MCP shell from *Haliangium ochraceum* forms a structure with $T=9$ icosahedral symmetry that exhibits 30° angles between its hexameric and pentameric components, which leads to a relatively spherical compartment shape. It is possible that the higher bending angle reported here imparts sharper angularity into the final Pdu MCP structure, consistent with previously reported cryo-TEM images of these structures. To this point we have added the following text to the manuscript, at lines 197-199:*

*“Notably, this is higher than bending angles (30°) between hexamer/pentamer components reported in the crystal structure of an MCP shell from *Haliangium ochraceum*.”*

To the second point, this calculation of bending angle should be considered as one component of the overall bending energy of the MCP shell. To extract the total bending energy for any given shell shape one would need to sum the energies associated with every interface. Thus, the angles observed would certainly be impacted by the presence of other hexamers (a consideration not included in our calculations). However, consideration of a PduA/PduA/PduN interface is instructive in how these pairwise interactions can be useful. In such a interface, all three bending angles (1 PduA/PduA interface, 2 PduA/PduN interfaces) would need to optimize simultaneously. Because the PduA/PduN interface has a strong energetic preference for a bent angle, both these interfaces will likely adopt a 40° interface. This would necessarily force the PduA/PduA interface to adopt a 40° angle as well, which strays from the 34° minimum observed in the PduA/PduA angle energy landscape; however, this change would come with a smaller energy penalty than changing the angle between the PduA/PduN interface.

The experimental characterization of the growth rate and metabolism shows that, while unbroken MCPs are important for growth and performance, Δ PduN has modest effects. A systems-level kinetic model is used to study the impact of the geometry of the microcompartment on the metabolic activity of the mutant cells. The model itself has been described in detail elsewhere; it makes reasonable assumptions about the kinetics of the reactions and assumes passive transport through perfectly closed shells. Experimentally, somewhat subtle changes in 1,2-propanediol are observed with the MT geometry (slightly slower consumption). In the systems model, this trend is reversed in the case of total volume and not obviously different from wild type when surface area is equated. So, while the geometry does indeed affect the results, it does so in a way that is consistent with the experimental observation of propionaldehyde production but inconsistent with 1,2-propanediol consumption. I think this point should be clearly addressed in the main text; currently there is no discussion of this point in the main text nor the supplement.

We thank the reviewer for noting this nuanced difference in the data and apologize for not noting it in the text. We agree that 1,2-propanediol consumption is slowed in the experimental data for the MT case, which is a feature not captured by either of the two limiting cases explored in the model. Based on this observation, we can conclude that this feature of the experimental data is not explained by changes in geometry described in the model and thus must be the result of some other phenomenon. To this point, we have added the following to lines 354-362 in the text:

“The experimental Pdu MT data show a slightly more rapid buildup of propionaldehyde in the Δ PduN strain than the wild type strain, suggesting that the overall surface area of these MTs may be slightly higher than MCPs. This conflicts with the 1,2-propanediol consumption data, in which the Δ PduN strain consumes 1,2-propanediol more slowly than the wild type strain, which is the opposite trend observed when surface area is increased in the model. Combined, this indicates that the geometry changes explored in the model cannot completely explain our experimental results; however, the modeling results do clearly describe how compartment geometry can impact encapsulated pathway kinetics.”

I find the explanation of the eventual consumption of the propionate and 1-propanol to be adequate. The fact, on the other hand, that propionaldehyde concentration is off by an order of magnitude requires a more serious discussion. If the subtle changes described in the supplement about temperature and shaking can affect concentration by that amount, the authors should make this clearer.

We thank the reviewer for noting their concern on this point. Experiments performed to evaluate the degree of propionaldehyde evaporation (data not shown) indicate that evaporative losses on the order of 50% can occur over a 12-hour window with shaking at 37 °C, as in the manuscript. The other factor not mentioned in the original version of the manuscript is that the propionaldehyde intermediate can also be acted upon by other enzymes in the cell, in part due to the generally high reactivity of aldehydes. We thus expect that the remainder of the discrepancies observed in the model result from consumption of the aldehyde intermediate by central metabolism. We have added discussion of this second point to the discussion in the supplement (Supplementary Discussion 1: Systems-level kinetic model, Discrepancies between model and experimental data).

The last point about the calculation that I do not quite understand: the cylindrical MT structures cannot be fully enclosed without pentamers. This means that the observed structure should be expected to be more like a straw than an enclosed cylinder. If diffusion of small molecules within the MCP is rapid, how does this structure provide the necessary passive barrier?

We thank the reviewer for this insightful question—it prompted analysis that we believe further illustrates the value of the kinetic modeling in hypothesis generation. We agree with the reviewer that the ends of the MT structures, due to geometric constraints, cannot be fully enclosed; however, the limitations of the structural characterizations described in this manuscript, particularly in vivo, preclude our ability to determine whether there is a diffusive barrier of some kind at the ends of the MTs. This is, however, an instance where our kinetic model can provide some insight, because we can examine the consequence of what occurs when changing the diffusivity of the MT ends. To this end, we constructed a modified model for

our MT system in which we allowed substrate diffusion out MT ends with a permeability set separately from the permeability of the MT axis. We found that permitting free diffusion at the MT ends in our model resulted in large propionaldehyde buildup. In fact, when we screened different permeabilities at the MT ends, we found that there needed to be a substantial diffusive barrier at the ends (corresponding to a permeability of 10^{-6} m/s) was required to substantially reduce this buildup. Thus, we conclude that something, whether it be excess shell protein aggregate or other macromolecular aggregates localized to cell ends where MT ends typically are, must be providing some degree of diffusive barrier out of our MT structures. We have added a description of the modified model to Supplementary Document 2, provided code for this model on GitHub, and have provided a detailed discussion of these results in Supplementary Document 1, including a figure (Supplementary Figure S9).

On the whole, while I am convinced that the authors have found a robust method for altering the geometry of the Pdu MCPs, the metabolic consequences of this change appear minor (one data point of 1,2-propanediol and propionaldehyde concentration differ outside error bars in Fig. 3 c). Rather than provide a clear mechanistic explanation of these small changes, the systems-level model indicates that a purely geometric explanation leads to changes that are qualitatively distinct from those in the experiment. The paradigm of modulating geometry of subcellular structures with the goal of impacting behavior, however, is an important, timely, and significant one.

We thank the reviewer for their careful review of our data and manuscript, and for appreciating the significance of these research directions in the MCP field.

Reviewer #2 (Remarks to the Author):

The authors report a comprehensive study on the interplay between structure and function of microbes microcompartments (MCPs) and how this knowledge can be used to characterize and leverage this sub-cellular organization. The study centers on pdu MCP as a model system, and specifically characterizes the role of two protein aggregates in the overall morphology of the compartment. PduN has a pentameric structure and it is known to form the vertices in encapsulated MCPs in *Salmonella enterica*. Whereas PduA has a hexameric spatial arrangement, and was used to model the majority of the “planar” constituents of the MCP at hand. The manuscript presents a systematic approach to study the molecular level details of PduN on the morphology of the compartment using experimental and computational approaches to differentiate encapsulated MCPs vs microtubules. The amount of mutants considered for the study is impressive and attests to the determination of the team to characterize the molecular factors that modulate microcompartment morphology and “composition.”

The manuscript as a whole is very easy to follow without taking away the technical aspects and scientific value of the study. The results are presented for the model system as well as the potential applications and broader impact in nanomaterials design, complex macromolecular assemblies (such as viral capsids), and modulation of microcompartment encapsulation metabolic profile/signature. One of the main outcomes presented by the authors is the use of specific techniques in (fluorescence) microscopy to track and characterize protein oligomers in the different encapsulated morphologies. This study is of interest to the wider scientific community and I recommend it for publication with minor reviews listed below.

1. The methods are all thoroughly described in the main manuscript and supporting documents. However, there are details missing about the computational approach; in line 686 the authors state more details about the MD simulations are provided in the supplement, but there are no such details in either Sup Doc 1 or 2. The authors should provide the following information to ensure reproducibility:

We thank the reviewer for noting these omissions, which are essential for reproduction of the work presented. Additional details regarding the simulations have been added to the manuscript, as described in detail below.

a. Dynamic ensemble and force field; any other pertinent settings to run the simulation (integration algorithms)

We have added the requested details to Supplementary Document 1 under “Supplementary Methods: All-atom molecular dynamics for calculation of bending potential, Simulation details”

b. Production simulation time (beyond initial relaxation of 100ps)

In Supplementary Document 1, under “Supplementary Methods: All-atom molecular dynamics for calculation of bending potential, Calculation of bending potential,” we now note that creating all of the independent windows for the calculations at different z-values takes about 20-30 ns pulling at 1 Angstrom/ns. In Supplementary Document 1, under “Supplementary Methods: All-atom molecular dynamics for calculation of bending potential, Calculation specifics,” we indicate the length of time required per window and the number of windows used in each calculation.

c. Number of windows used to compute the PMF and details for umbrella sampling

We have added to Supplementary Document 1, under “Supplementary Methods: All-atom molecular dynamics for calculation of bending potential, Calculation of bending potential,” a note that the harmonic bond constant, k, for umbrella sampling in these simulations is 1000 kJ/mol. In Supplementary Document 1, under “Supplementary Methods: All-atom molecular dynamics for calculation of bending potential, Calculation specifics,” we provide the length of time simulated per window and the number of windows used.

d. Assess/discuss convergence of the PMFs presented in figure 3.

In Supplementary Document 1, under “Supplementary Methods: All-atom molecular dynamics for calculation of bending potential, Calculation specifics,” we note that error bars on data points are calculated by dividing up the data into subsections (i.e. comparing different time sections of the data, comparing odd time points versus even time points) and running the calculation on these smaller portions of the data to estimate error bars on sampling. These sampling error bars, we believe, provide evidence of convergence, as simulations were run until these sampling error bars were deemed sufficiently smaller than the characteristic energetic wells we were investigating. For example, a 2 kcal/mol error on the 6 kcal well was deemed sufficient for the bending of the PduN-PduA interface. However, for the bending of the PduA-PduA interface, which had only a well of 1.2 kcal/mol, we ran simulations for longer times to reduce error to 0.5 kcal/mol, to establish confidence that the energy minimum in this landscape was statistically significant.

e. Were any replicas run for the trajectories?

As described in the response to point d above, the manner in which we subdivided our trajectories to estimate error in our PMF essentially treats the different sections of time in the

dataset as replicates for the purpose of calculating error; however, multiple simulations using they same initial conditions were not run.

2. The section labeled “discussion” should be Conclusions or Outlook. The authors provide excellent context and evidence for their hypotheses and conclusions in the Results section, which in fact can perfectly be labeled as “Results and discussion.”

We thank the reviewer for their support! We have adjusted the titles accordingly (Lines 105 and 481).

Reviewer #3 (Remarks to the Author):

The authors present a very clear manuscript describing the impact of changes to the PduN vertex protein on the morphology of the Salmonella enteric Pdu micro compartment. Elucidating the structure and function of bacterial metabolosomes is of fundamental importance in understanding the role of these in the metabolism of niche carbon sources, particularly in enteric pathogens like Salmonella enterica. Further to this basic research, metabolosomes present biotechnological important 'reaction' vessels for metabolic engineering.

The results presented by the authors demonstrate the role of the PduN vertex protein in determining the closure of an icosahedral metabolosome, and in its absence the formation of tubular structures.

This manuscript is very well written and presents important results and new methodological approaches for screening variant metabolosomes microscopically. I would recommend publication with only minor changes for clarity.

In terms of clarifications/additions to enhance the manuscript I have the following minor comments to the authors.

Figure 2 - panel b - note the addition of lysozyme in the figure legend (I am assuming the band indicated 'Lys' is lysozyme. In panels C/D it is very hard to identify the GFP fluorescence indicated in the micrographs. Can this figure be adjusted in some way to enhance the contrast/signal from these micrographs?

We thank the reviewer for pointing out this potential confusion in the figure. We have added a description of what “Lys” indicates to the legend of Figure 2 and adjusted the contrast in GFP micrographs to improve visibility of the fluorescent structures.

Figure 4: Panel b - can the specific growth rates for the various mutants be calculated to give solid values to the differences in growth for the mutants? From a purely visual estimation the exponential growth rates are very similar, but it would be good to have a number to compare these.

Per the reviewer’s suggestion, we calculated doubling times for the growth curves in the regions described in the text and added these numbers to the text at lines 249-250, 253-254, 262-263, and 265-267.

Figure 5 - the assumption that there is no significant diffusion of substrates across the MT ends is not clear from my reading of the manuscript, or the supplementary material. Could you be more explicit in your reasoning for this assumption. Do you suggest that the MT ends have no

shell proteins, or are they capped in some way? Either way one would imagine that there would be some diffusion across this area.

We agree with the reviewer that this is an important question surrounding these structures; per the sixth response to Reviewer 1 (see above), we have added modeling results and further discussion to the submission to address this point.

Figure 7 - GFP panels do not have sufficient contrast.

We again thank the reviewer for noting that these images did not provide sufficient contrast for independent reader interpretation. We have increased the contrast accordingly.

Line 489 - repetition of 'towards a fundamental understanding' leads to a lack of clarity of your message. Can you reword for clarity?

We thank the reviewer for noting this unclear phrase in the manuscript. Per the reviewer's suggestion, we have edited the noted sentence as follows (lines 511-513):

"In the context of understanding MCP assembly, this enables rapid characterization of the fitness and mutability of different interacting residues in shell proteins like PduN and PduA."

Methods - line 524 - can you give a reference to the TUC01.

References to the appropriate sources have been added to the main text as requested (Line 548).

Reviewer #4 (Remarks to the Author):

This manuscript presents a genetic, metabolomics, and computational modeling analysis of the role played by the pentameric shell protein PduN in the pdu microcompartment. Experimental data show that deletion of the pentamer tends to give elongated tubular structures, sometimes interfering with complete cell division. Despite the structural defects, the aberrant containers appear to offer reasonably good metabolic performance compared to complete absence of microcompartments, though with some more nuanced differences compared to wild-type in terms of product balance, which could be directly related to biochemical defects or to secondary mechanisms. The study is supported by diffusion and kinetic modeling. Similarly, the findings there show some dependence on shape and size parameters rather than major deviations compared to containers of wild-type shape. A weakness of the modeling approach is that it appears the openness of the tubes could not be modeled effectively with the approach taken and it is hard to know if diffusion through open ends would have had an important effect in a computational approach where that could be accounted for.

We agree with the reviewer that this is an important question surrounding these structures; per the sixth response to Reviewer 1 (see above), we have added modeling results and further discussion to the submission to address this point.

There is also a molecular dynamics analysis of the energetics of association between a shell protein hexamer and a pentamer, focusing on bending energetics. This part of the study, which is a bit tangential to the main gene function and metabolic aspects of the paper, appears to have a problem with the mathematical treatment of energetics vs bending angle, which the

authors will need to revisit. Overall, the work brings a combination of sophisticated tools to bear on some open questions about function in bacterial microcompartments. The findings are not too surprising, but sensible and worth publishing in a good venue. Much of the value of the study could be in its approaches, including computational methods, which might find use in other complex systems. With that in mind, I think the work could have positive impact if technical issues (below) can be ironed out.

Major point:

There is some complexity in how the authors have sought to analyze the energetics of hexamer/pentamer bending on the basis of MD simulations carried out over a series of starting conformations at a range of bending angles. The description of how those calculations were analyzed is tough to follow, but after a few passes I was able to come (I think) to an understanding. Based on that, my conclusion is that an important mathematical error was made in going from the analysis of local MD results (as a function of z) to what is supposed to amount to a kind of path integral over the bending angle θ . The key equation, around line 15-16 in the supplement, would work if the energy V was dependent on z and not any other (rigid body) variable. But that is not the case. In order to get a correct treatment of V by integrating approximate values of dV as θ changes, the authors need to start with the complete differential of V with respect to the rigid body positions. If it's reasonable to treat that as a z component and a radial component (in the y direction, as the authors have set things up), then the starting point would be

$$dV/d(\theta) = dV/dz dz/d(\theta) + dV/dy dy/d(\theta)$$

In their treatment, the authors have left out the second term. I believe that after thinking it through the authors will agree that it is not possible to obtain a valid measure of the difference in V between an initial and final value of θ by only considering the partial derivative of V with respect to z , but not y . It is easy to construct a trivial scenario to show this. E.g. if a function V varies as a function of y but is independent of z , which though not likely is entirely permissible mathematically, then the authors' equation (using only the first term), would predict no change in V from the beginning to ending values of θ , even though y (and hence V) would have changed. If the authors are to rescue their attempt to obtain V as a function of θ , then they would need to reexamine their MD data to extract estimates of dV/dy . Perhaps the numerical effects of including the dependence on y will be modest. It is hard to be sure. But the treatment does require that term. I spent quite a bit of time coming to grips with the treatment, so I am 99% confident of the conclusion here, unless somehow the authors can show what happens to the missing term in the total derivative. [There doesn't seem to be any reason to think it would be much smaller than the term that was included.]

We apologize to the reviewer for this point of confusion and thank them for taking the time to understand and evaluate this approach—in our original version of the manuscript, we omitted details required for accurately interpreting the approach. To ameliorate this issue, we have made significant revisions to Supplementary Document 1 as well as the Materials and Methods

in the main text (in the second paragraph of the “All-atom molecular dynamics simulations” section).

In summary, we clarified that forces reported were sampled using umbrella sampling. In this technique, a harmonic spring constrains the distance between the hexamer and the pentamer (or the two hexamers) in the z direction, and we measure the average force required to maintain this constraint. The $\cos(\theta)$ term provides the component of this force in the θ direction. In the original manuscript, we neglected the dV/dy term indicated by the reviewer because the only force applied in this umbrella sampling approach was, in fact, in the z-direction, rendering the force in the y-direction, dV/dy , zero. As the reviewer correctly noted, this was inconsistent with our original definition of V , and we believe this inconsistency is the source of the confusion indicated by the reviewer. Clarification of this point has been added to Supplementary Document 1, Supplementary Methods: All-atom molecular dynamics for calculation of bending potential, Calculation of bending potential. We have also added details to the main text in the Methods section under “All-atom molecular dynamics simulations.”

Minor comment:

In the same equation in the supplement: in the end I suppose it is cosmetic, but the equation as written seems to introduce some unit problems in the intermediate terms (which then cancel). As written, $dV/d(\theta)$ would not agree with the units for a Newtonian force. It seems that a dependence on radius should probably be introduced formally.

F should probably be written as $= -(1/r) dV/d(\theta)$. And in the next step, $dz/d(\theta)$ should be $r \cdot \cos(\theta)$. The $(1/r)$ and r then cancel.

[As noted above, it seems a correct treatment would require terms for dependence on y , treated similarly.]

We thank the reviewer for indicating this error. After making the changes described in the previous response to address the reviewer’s concerns, this equation no longer exists in the text.

References

1. Tanaka, S. *et al.* Atomic-Level Models of the Bacterial Carboxysome Shell. *Science* **319**, 1083–1086 (2008).
2. Sutter, M., Greber, B., Aussignargues, C. & Kerfeld, C. A. Assembly principles and structure of a 6.5-MDa bacterial microcompartment shell. *Science* **356**, 1293–1297 (2017).
3. Sutter, M., Wilson, S. C., Deutsch, S. & Kerfeld, C. A. Two new high-resolution crystal structures of carboxysome pentamer proteins reveal high structural conservation of CcmL orthologs among distantly related cyanobacterial species. *Photosynth Res* **118**, 9–16 (2013).
4. Mallette, E. & Kimber, M. S. A Complete Structural Inventory of the Mycobacterial Microcompartment Shell Proteins Constrains Models of Global Architecture and Transport. *J Biol Chem* **292**, 1197–1210 (2017).
5. Wheatley, N. M., Gidaniyan, S. D., Liu, Y., Cascio, D. & Yeates, T. O. Bacterial microcompartment shells of diverse functional types possess pentameric vertex proteins. *Protein Science* **22**, 660–665 (2013).
6. Keeling, T. J., Samborska, B., Demers, R. W. & Kimber, M. S. Interactions and structural variability of β -carboxysomal shell protein CcmL. *Photosynth Res* **121**, 125–133 (2014).

7. Greber, B. J., Sutter, M. & Kerfeld, C. A. The Plasticity of Molecular Interactions Governs Bacterial Microcompartment Shell Assembly. *Structure* **27**, 749-763.e4 (2019).

Reviewers' Comments:

Reviewer #1:

Remarks to the Author:

I appreciate the thorough effort to address my comments; the authors have clarified many aspects of their interpretation of the data and the comparison with the model. While I think the model and data continue to have some notably discrepancies, I feel that these are appropriately contextualized. I recommend publication.

Reviewer #2:

Remarks to the Author:

The authors responded all my comments and concerns. In general, the authors have addressed well the points submitted by the reviewers. I support the publication of this manuscript

Reviewer #3:

Remarks to the Author:

We thank the authors for their thorough consideration of the review points and they present a strengthened and clarified manuscript. This is an interesting study and should be published at this point as is.

Reviewer #4:

Remarks to the Author:

The authors have made suitable revisions to the manuscript. On the whole, the study is rather complex, and the modeling doesn't fully capture all the observed metabolic behaviors. But credit to the authors for a thorough and expert analysis of these still-incompletely understood systems, including features that are not yet within the scope of modeling. The manuscript appears ready for publication.

REVIEWERS' COMMENTS

Reviewer #1 (Remarks to the Author):

I appreciate the thorough effort to address my comments; the authors have clarified many aspects of their interpretation of the data and the comparison with the model. While I think the model and data continue to have some notably discrepancies, I feel that these are appropriately contextualized. I recommend publication.

We thank the reviewer for their recommendation for publication.

Reviewer #2 (Remarks to the Author):

The authors responded all my comments and concerns. In general, the authors have addressed well the points submitted by the reviewers. I support the publication of this manuscript
We thank the reviewer for their support of publication.

Reviewer #3 (Remarks to the Author):

We thank the authors for their thorough consideration of the review points and they present a strengthened and clarified manuscript. This is an interesting study and should be published at this point as is.

We thank the reviewer for the support of this work.

Reviewer #4 (Remarks to the Author):

The authors have made suitable revisions to the manuscript. On the whole, the study is rather complex, and the modeling doesn't fully capture all the observed metabolic behaviors. But credit to the authors for a thorough and expert analysis of these still-incompletely understood systems, including features that are not yet within the scope of modeling. The manuscript appears ready for publication.

We thank the reviewer for acknowledging the complexity of the system and their recommendation for publication.